# *Candida albicans* translocation through the intestinal epithelial barrier is promoted by fungal zinc acquisition and limited by NFκB-mediated barrier protection

Jakob L. Sprague[1], Tim B. Schille[1,2], Stefanie Allert[1], Verena Trümper[1], Adrian Lier[1], Peter Großmann[3], Emily L. Priest[4], Antzela Tsavou[4], Gianni Panagiotou[2,3,5], Julian R. Naglik[4], Duncan Wilson[6], Sascha Schäuble[3], Lydia Kasper[1☯], Bernhard Hube[1,2,5☯]*

1 Department of Microbial Pathogenicity Mechanisms, Hans-Knöll-Institute, Jena, Germany, 2 Cluster of Excellence Balance of the Microverse, Friedrich-Schiller-University Jena, Jena, Germany, 3 Department of Microbiome Dynamics, Hans-Knöll-Institute, Jena, Germany, 4 Centre for Host-Microbiome Interactions, Faculty of Dentistry, Oral and Craniofacial Sciences, King's College London, London, United Kingdom, 5 Institute of Microbiology, Friedrich-Schiller-University Jena, Jena, Germany, 6 Medical Research Council, Centre for Medical Mycology at the University of Exeter, Exeter, United Kingdom

☯ These authors contributed equally to this work.
* bernhard.hube@leibniz-hki.de

**Data Availability Statement:** Programming code and data necessary to generate plots shown in this manuscript were deposited at Github: https://

## Abstract

The opportunistic fungal pathogen *Candida albicans* thrives on human mucosal surfaces as a harmless commensal, but frequently causes infections under certain predisposing conditions. Translocation across the intestinal barrier into the bloodstream by intestine-colonizing *C. albicans* cells serves as the main source of disseminated candidiasis. However, the host and microbial mechanisms behind this process remain unclear. In this study we identified fungal and host factors specifically involved in infection of intestinal epithelial cells (IECs) using dual-RNA sequencing. Our data suggest that host-cell damage mediated by the peptide toxin candidalysin-encoding gene *ECE1* facilitates fungal zinc acquisition. This in turn is crucial for the full virulence potential of *C. albicans* during infection. IECs in turn exhibit a filamentation- and damage-specific response to *C. albicans* infection, including NFκB, MAPK, and TNF signaling. NFκB activation by IECs limits candidalysin-mediated host-cell damage and mediates maintenance of the intestinal barrier and cell-cell junctions to further restrict fungal translocation. This is the first study to show that candidalysin-mediated damage is necessary for *C. albicans* nutrient acquisition during infection and to explain how IECs counteract damage and limit fungal translocation via NFκB-mediated maintenance of the intestinal barrier.

## Author summary

*Candida albicans* populations colonizing the intestine serve as the main source for systemic infections. Though normally commensal, under certain conditions, *C. albicans* can

github.com/SchSascha/Cal_Translocation. Raw sequencing data is submitted under project accession number GSE237496 to the GEO gene accession omnibus.

**Funding:** JLS, AL, LK, BH, SaS, PG, and GP were supported by the German Research Foundation (Deutsche Forschungsgemeinschaft – DFG) within the Collaborative Research Centre (CRC)/ Transregio (TRR) 124 "FungiNet" projects C1 and INF (DFG project number 210879364). TBS was supported by the DFG under Germany's Excellence Strategy – EXC 2051 – Project ID 390713860. VT was supported by the German Federal Ministry of Education and Research (BMBF) within the funding program Photonics Research Germany, Leibniz Center for Photonics in Infection Research (LPI) (subproject LPI-BT2; contract number 13N15705). GP was also supported by the BMBF within the funding project PerMiCCion (project ID: 01KD2101A). JRN was supported by grants from the Wellcome Trust (214229_Z_18_Z) and National Institutes of Health (DE022550). DW was supported by a Wellcome Trust Senior Research Fellowship (214317/Z/18/Z), the MRC Centre for Medical Mycology at the University of Exeter (MR/N006364/2 and MR/V033417/1), and the NIHR Exeter Biomedical Research Centre. The views expressed are those of the author(s) and not necessarily those of the NIHR or the Department of Health and Social Care. The funders had no role in study design, data collection and analysis, decision to publish, or preparation of the manuscript.

**Competing interests:** The authors have declared that no competing interests exist.

translocate across the intestine and into the bloodstream, leading to systemic candidiasis. Here we dissect the fungal and host activities involved in this process. We find that damage to host cells, which supports efficient translocation, is associated with active acquisition of host-cell zinc by *C. albicans*. At the same time, intestinal epithelial cells foster barrier integrity to limit fungal translocation independently of host damage.

## Introduction

The majority of life-threatening invasive *Candida* infections are caused by *C. albicans* [1–3]. The WHO have acknowledged the global threat posed by fungal pathogens and recently published a priority list which included *C. albicans* in the critical priority group [4]. *C. albicans* normally exists as a commensal member of the mycobiota. Within a healthy host the resident microbiota, epithelial barriers, and the host immune system keep *C. albicans* commensal and prevent translocation over the intestinal barrier into the blood stream [5–7]. The fungus can, however, transition to a pathogenic state under certain predisposing conditions [5]. In fact, systemic infections arise from endogenous *C. albicans* populations within the gastrointestinal (GI) tract and require translocation across the intestinal epithelium into the bloodstream [6,8–10]. Translocation events occur in patients suffering from a variety of predisposing conditions, such as long-term use of broad-spectrum antibiotics or immunosuppression, and the resulting infections are challenging to diagnose, difficult to treat effectively, and associated with high mortality rates [2]. However, the specific fungal factors and host mechanisms that contribute to fungal translocation still remain largely undefined.

In an *in vitro* model of cultured intestinal epithelial cells, *C. albicans*-mediated damage due to the hypha-associated fungal cytolytic toxin candidalysin was described as a major mechanism of fungal translocation *via* a transcellular route [11]. In line with this, fungal genes that are necessary for hyphal growth or delivery of candidalysin were needed for the full translocation capacity [11,12]. Nevertheless, low-level translocation was possible in strains lacking candidalysin, suggesting that further, yet uncharacterized factors and mechanisms contribute to translocation.

The host response to *C. albicans* infection has been extensively described in oral epithelial cells (OEC). *C. albicans* infection activates NFκB and c-Jun-based MAPK signaling pathways in OECs independent of fungal morphology [13]. A second MAPK signaling phase involving MKP1 and c-Fos is triggered by the activation of epidermal growth factor receptor (EGFR) by candidalysin-mediated host-cell damage [13–15]. This innate immune response results in the production of cytokines like IL-1β or IL-8 [15,16]. Independent of NFκB and MAPK signaling, *C. albicans* also induces PI3K/Akt signaling during infection of OECs [17]. The response of IECs to *C. albicans* infection is less well characterized. IECs respond *via* MAPK and TNF signaling as well as activation of NFκB which protects from fungal-mediated host-cell damage [18]. Dual-species RNA sequencing has proven to be a powerful tool for identifying the time-resolved infection-specific activities of *C. albicans* and host cells during their interaction [19–22].

In this study, we therefore used dual-species transcriptional profiling to investigate the molecular dynamics upon interaction of *C. albicans* with IECs and to determine which fungal and epithelial processes contribute to IEC damage and fungal translocation.

We found that the candidalysin-encoding gene *ECE1* is required for zinc acquisition during invasion of host cells. This indicates that *C. albicans*-induced host-cell damage supports the acquisition of host micronutrients, in this case zinc, during infection. We also show that *ECE1*-dependent host damage and subsequent fungal translocation are limited by NFκB-

mediated maintenance of the epithelial barrier. Consequently, NFκB activation by IECs limits the pathogenic potential of *C. albicans* and helps to protect epithelial barrier integrity.

## Results

### Transcriptional adaptation of *C. albicans* to IEC infection includes dynamic metabolic shifts and indicates scavenging of host zinc

To characterize the interaction of *C. albicans* with intestinal epithelial cells (IECs) from early fungal adhesion to initial invasion (up to 6 h) and later translocation and damage phases (12–24 h), we conducted dual-species RNA sequencing of *C. albicans*-infected IECs over a 24 h time course. Intestinal epithelial cells (C2BBe1) were seeded and differentiated for 12 d after reaching confluency to form a polarized cell layer [23]. Fungal and host RNA was isolated from *C. albicans* and IECs alone as well as from co-incubated samples. Samples were taken before infection (0 h), at 45 min (0.75 h), 3 h, 6 h, 12 h, and 24 h. *C. albicans* showed a clear time-dependent transcriptional response with distinct clusters for each time point (Fig 1A). *C. albicans* samples cultured with and without IECs showed similar transcriptional responses and clustered together, indicating the fungal response in this experimental model was largely independent of the presence of host cells (Fig 1A).

To explore the transcriptional changes in *C. albicans* that are specific for the presence of IECs, we identified infection-specific differentially expressed genes (DEGs). For that, we compared fungal transcript levels in the presence *vs* absence of IECs at each respective time point (S1 Table). Using these DEGs, we performed functional enrichment analysis of gene ontology terms (GO; biological process category; S2 Table and Fig 1B) to identify the molecular patterns of infection.

We first looked at stress- or virulence-related functional categories that were enriched upon infection. In the earliest stages of infection, at 45 min, the enrichment analysis detected the term "response to chemical" which contained the stress-related catalase-encoding gene, *CAT1*, and the thioredoxin-encoding gene, *TRR1* (S2 Table and Fig 1B). Decreased mRNA levels of these genes in infected samples compared to medium-only controls (S1 Table) suggests that *C. albicans* does not experience significant stress upon early contact to IECs. In late infection stages, however, the category "detoxification" was significantly overrepresented among infection-specific DEGs with increased transcript levels during infection (Fig 1B). These included many genes encoding predicted membrane transporters like the putative ABC transporter, *CDR4*, and the putative MFS transporter, *QDR3* (S2 Table). This indicates increased stress related to the host or to host-cell damage.

The majority of enriched GO terms did not contain classical virulence-associated categories and did not include genes related to fungal filamentation or virulence. Only few enriched terms, including some related to lipid metabolism and cell adhesion, fall into these categories. Genes involved in lipid metabolism were significantly overrepresented at 6 h among the infection-specific DEGs (Fig 1B). These included many genes needed for ergosterol biosynthesis, which showed increased expression during infection of IECs (S1 and S2 Tables). As ergosterol biosynthesis is important for filamentation, this increased expression may indicate infection-specific surface changes in invading hyphae [24]. Enrichment of the category "cell adhesion" at 12 h similarly points to infection-specific surface changes (Fig 1B). This category includes DEGs coding for adhesins, cell surface proteins and cell wall integrity-related genes, such as *EAP1*, *ALS1*, *PRA1*, *HWP1*, *XOG1*, *MNT2* and *PMT1* (S1 and S2 Tables).

Most infection-specific changes were, however, due to metabolic adaptations of *C. albicans*. At time points of initial hypha formation prior to substantial host cell invasion (3 h) but also during host cell invasion and damage (12 h and 24 h) [11], the enrichment analysis revealed

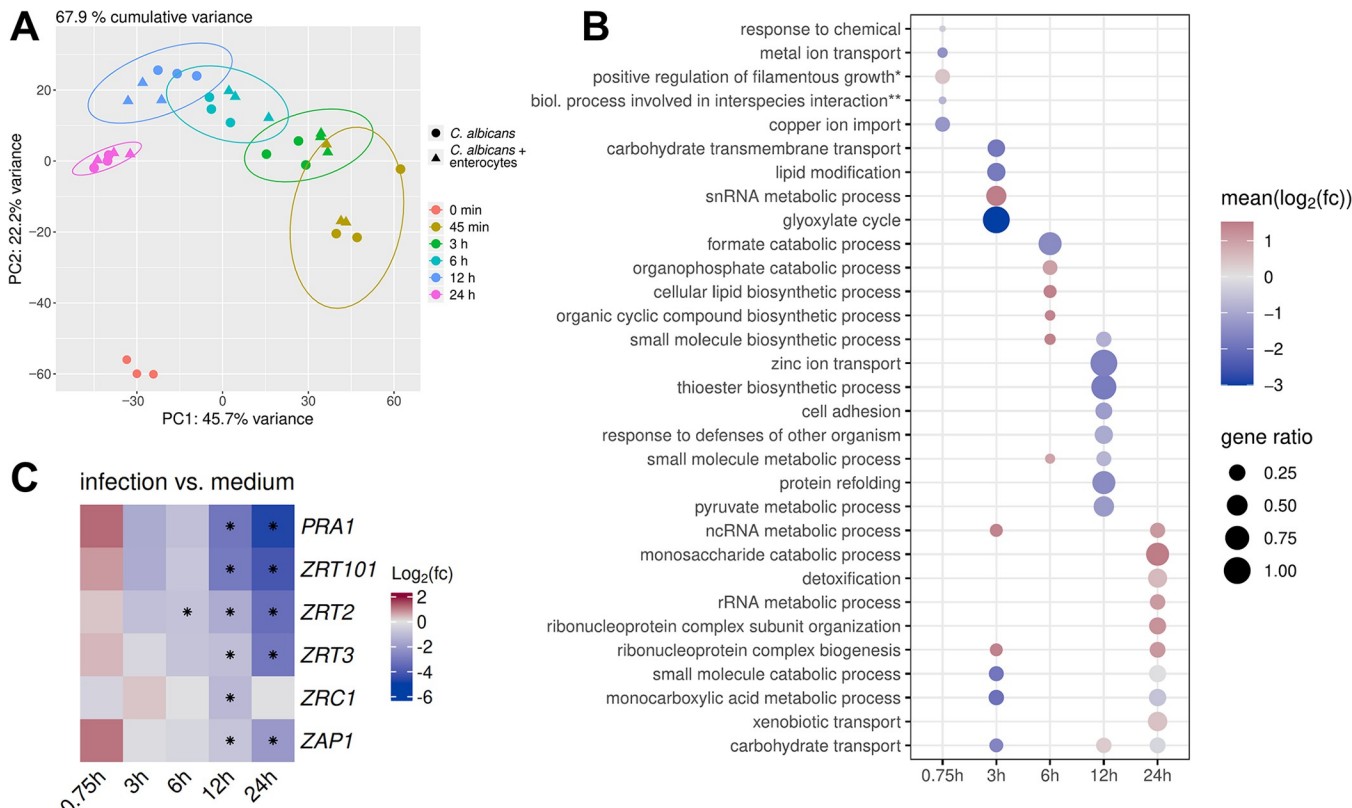

**Fig 1. *C. albicans* shows transcriptional changes specific to the presence of IECs during infection.** (A) Principal component analysis for WT (SC5314) *C. albicans* RNA sequencing data. Samples clustered together depending on the time point regardless of the presence of IECs (enterocytes). (B) GO terms significantly overrepresented among infection-specific *C. albicans* DEGs (*positive regulation of filamentous growth of a population of unicellular organisms in response to chemical stimulus; **biological process involved in interspecies interaction between organisms). Gene ratio represents the proportion of genes within each category that were significantly differentially expressed during infection compared to *C. albicans* cells only. The color scale represents the mean Log₂(fold-change) of significant DEGs within each category. Red indicates that most DEGs showed increased expression during infection, blue indicates most DEGs showed decreased expression, and grey indicates a mix of DEGs with increased and decreased expression within the category. Dots are only shown for significantly enriched categories (overrepresentation analysis, Benjamin-Hochberg adjusted p-value ≤ 0.05, S2 Table). (C) Gene expression of zinc-related genes comparing *C. albicans* cell during infection of IECs to those cultured in medium only. The color scale represents the Log₂(fold-change) for each gene during infection of IECs compared to medium only, with red representing higher expression during infection and blue representing lower expression during infection. Genes for the zinc transporters *ZRT101*, *ZRT2*, *ZRT3* and *ZRC1*; the zinc-scavenging protein *PRA1*; and the transcription factor *ZAP1* (*CSR1*) were less expressed during infection of IECs. An asterisk indicates the time points with a statistically significant difference in gene expression (DESeq2 p < 0.05). The mean MRN values are given in S1 Table.

adaptations in the central fungal carbon metabolism in categories for carbohydrate transmembrane transport, the glyoxylate cycle, and monocarboxylic acid metabolism were enriched at 3 h (Fig 1B). These categories cover glucose transporter genes like *HGT2* and *HGT12*, which are known to be repressed by high glucose levels [25], and genes involved in utilization of non-glucose carbon sources *via* the glyoxylate cycle and fatty acid beta oxidation, like *ICL1*, *MLS1*, *FOX2*, and *PEX5* (S2 Table). Lower transcript levels of these genes during infection (S1 Table) indicates that the presence of IECs provides glucose to infecting *C. albicans* cells even before substantial fungal invasion. In addition, there was an infection-specific increase in translational activity at 3 h, with higher transcript levels of genes within the categories for ribonucleoprotein biogenesis and snRNA metabolic processes (Fig 1B). This indicates infection-specific re-organization of central cellular processes at an early time point.

At 12 h, genes falling under the term "pyruvate metabolic process" were also significantly enriched, including genes relevant for glycolysis (*ENO1*, *CDC19*, *TDH3*, and *FBA2*) and for

the pyruvate dehydrogenase complex (*PDA1*, *PDB1*, *PDX1*, and *LAT1*) (Fig 1B and S2 Table). The lower transcript levels of these genes points to reduced glucose availability during these later stages of IEC infection (Fig 1B and S1 Table). Finally, at 24 h, the term "monosaccharide catabolism" was overrepresented among the infection-specific DEGs. This suggests that non-glucose C6 carbon sources are available specifically during the host-cell damage phase of infection (Fig 1B). These genes included some involved in galactose metabolism (*GAL1*, *7*,*10*) and glycolysis (*PFK1*, *2*) which were more highly expressed in the presence of IECs (S1 and S2 Tables). Translation-related processes were again enriched during infection at the 24 h time point (Fig 1B). Potentially, this reflects an adaptation to the more complex environment when *C. albicans* gains access to the host cell content.

Taken together, our data indicate that *C. albicans* dynamically adapts to the high availability of its preferred carbon source glucose at 3 h, to glucose limitation at 12 h, and the availability of alternative carbon sources upon host-cell damage at 24 h.

Importantly, not only did our analysis pick up infection-specific changes in macronutrient metabolism, but also adaptation to micronutrient availability. At 12 h, the term "zinc ion transport" was significantly overrepresented among the infection-specific DEGs (Fig 1B). This term contained the zinc transporter genes *ZRT101*, *ZRT2*, *ZRT3*, and *ZRC1*, which all had significantly reduced transcript levels during IEC infection. The zincophore gene *PRA1* and the zinc homeostasis regulator gene *ZAP1* (*CSR1*) showed the same pattern (Fig 1C) [26]. While mRNA levels of these zinc-related genes increased over time in both infecting *C. albicans* and *C. albicans* in medium only, their levels were higher in the absence of IECs, especially at later stages of infection (S1 Fig). The presence of their substrates strongly down-regulates the transcription of genes involved in micronutrient acquisition, like those for zinc [27–29]. Together, our data therefore indicate a stronger starvation for zinc when *C. albicans* is grown in the absence of IECs. In contrast, other metal-related terms were not overrepresented except for copper ion transport at initial infection stages (45 min) (Fig 1B). In fact, increasing transcript levels of iron-metabolic genes, like the high-affinity iron permease genes *FTR1* and *FTR2*, in both infection and control conditions indicated a general iron starvation response at later time points. Unlike for zinc, this was not rescued by the presence of IECs (S1 Table).

Overall, our transcriptome analysis indicates that *C. albicans* adapts its metabolism and surface to the dynamic environmental changes during experimental IEC infection, including acquisition of zinc from the host cells.

## Zinc acquisition during infection of IECs is *ECE1*-dependent

Zinc ion transport was a significantly overrepresented category among infection-specific DEGs during the later stages of invasion and host-cell damage (Fig 1B). This was not the case for genes involved in the transport of other metals or micronutrients, so we hypothesized that zinc may be of particular importance. To determine whether zinc acquisition plays a role during *C. albicans* infection of IECs, we utilized gene deletion mutants for the zinc importer genes *ZRT101* and *ZRT2*, the zincophore gene *PRA1*, and the intracellular zinc transporter gene *ZRC1*. Zrt101, Zrt2, and Pra1 are involved in zinc acquisition and are upregulated by low zinc [27,28]; in contrast, Zrc1 is involved in zinc detoxification, although its transcriptional regulation has not been investigated. Loss of *ZRT101*, *ZRT2*, and *PRA1* had no impact on adhesion to, invasion of, or translocation through IECs by zinc-pre-starved *C. albicans* cells (Fig 2A–2C). The strain lacking *ZRC1*, however, showed decreased invasion and strongly impaired translocation (Fig 2B and 2C). The dramatic decrease in translocation ability (~10-fold) is unlikely to be solely due to the decreased hypha formation (~1.5-fold) of this strain (S2 Fig). In contrast to these results, we found

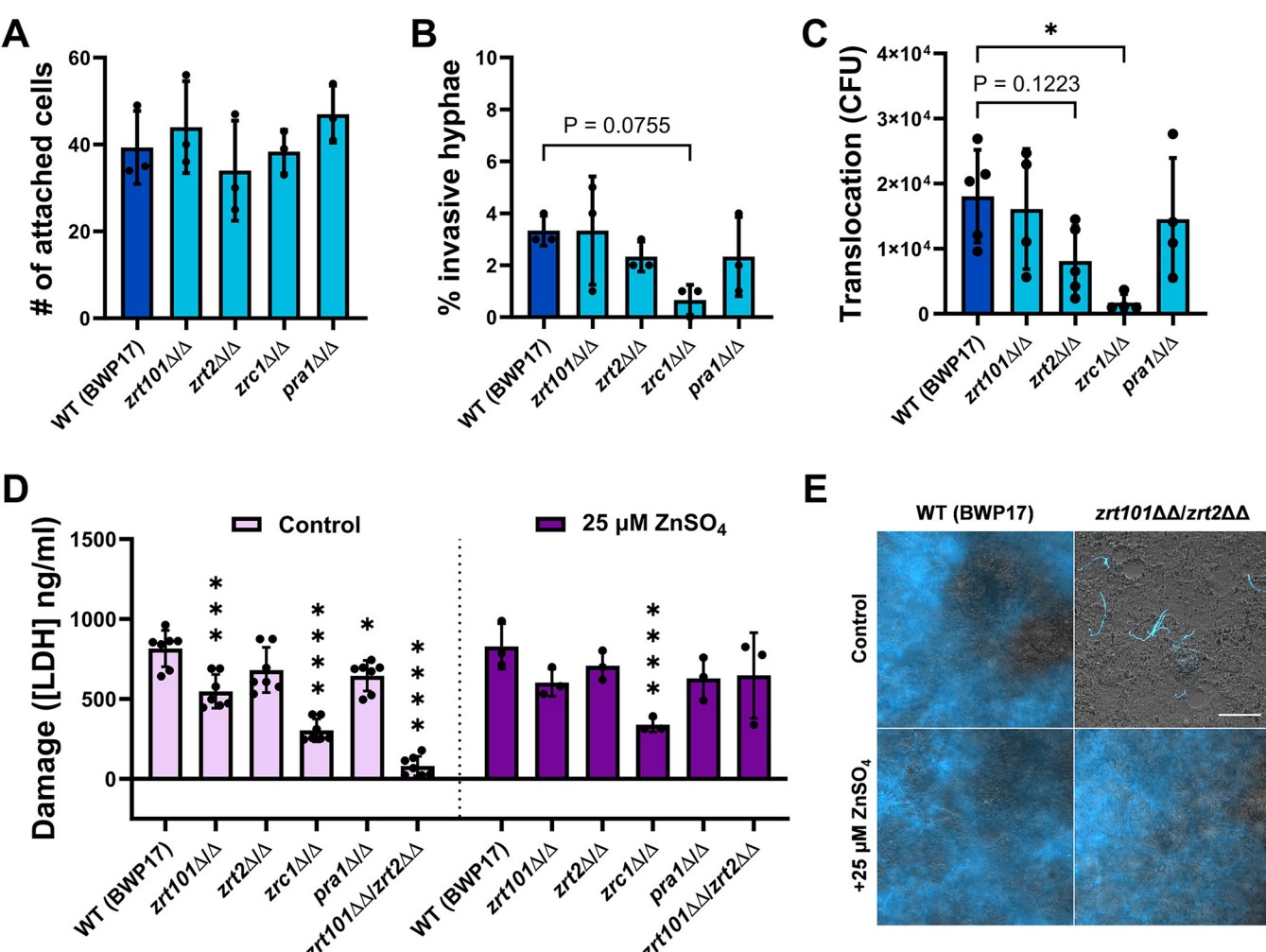

**Fig 2. Loss of zinc acquisition genes in *C. albicans* decreases damage potential by impairing growth during infection of IECs.** (A) Loss of zinc acquisition and storage genes had no effect on the adhesion of *C. albicans* to C2BBe1:HT29-MTX intestinal epithelial cells after 3 h. (B) Zinc acquisition and storage genes did not significantly affect the percentage of invasive hyphae after 6 h co-incubation with C2BBe1:HT29-MTX cells. The invasion of *zrc1*Δ/Δ was reduced, though not to a statistically significant degree. (C) Loss of *ZRC1* significantly impaired translocation of *C. albicans* cells through C2BBe1:HT29-MTX intestinal epithelial cells grown on a porous membrane after 24 h. (D) The host-cell damage of the *zrt101*Δ/Δ, *zrc1*Δ/Δ, *pra1*Δ/Δ, and *zrt101*ΔΔ/*zrt2*ΔΔ strains was significantly reduced after 24 h of infection. The severe damage impairment could be rescued with exogenous addition of 25 μM ZnSO₄ during infection for *zrt101*ΔΔ/*zrt2*ΔΔ ($P < 0.0001$), but not for *zrc1*Δ/Δ ($P > 0.9999$). LDH release was adjusted by subtracting the release from uninfected and untreated host cells. (E) The addition of exogenous zinc during infection of C2BBe1:HT29-MTX cells increases the fungal growth of the *zrt101*ΔΔ/*zrt2*ΔΔ strain to a level similar to that of the WT (BWP17) strain. Fungal hyphae (blue) were stained with calcofluor white (scale bar = 100 μm). All values are shown as the mean with standard deviation. Invasion (B) and translocation (C) data were compared using a one-way analysis of variance (ANOVA) and the host-cell damage (D) data were compared using a two-way ANOVA. Statistical significance was determined with a post-hoc Dunnett's multiple comparisons test. Statistical significance: *, $P \leq 0.05$; ***, $P \leq 0.001$; ****, $P \leq 0.0001$.

that loss of *ZRT101*, *PRA1*, and *ZRC1* results in significantly decreased damage potential, and an almost complete absence of host-cell damage after deletion of both *ZRT101* and *ZRT2* (Fig 2D). The host-cell damage defect of this zinc uptake impaired mutant (*zrt101*ΔΔ/*zrt2*ΔΔ), but not of the zinc detoxification defective mutant (*zrc1*Δ/Δ) was rescued by addition of 25 μM ZnSO₄, a zinc concentration known to promote *C. albicans* growth while not being toxic (Fig 2D) [28]. This restoration of host-cell damage is likely due to increased growth and filamentation as assessed by microscopy images showing that without addition of zinc there is little-to-no fungal growth for the *zrt101*ΔΔ/*zrt2*ΔΔ strain (Fig 2E). Therefore,

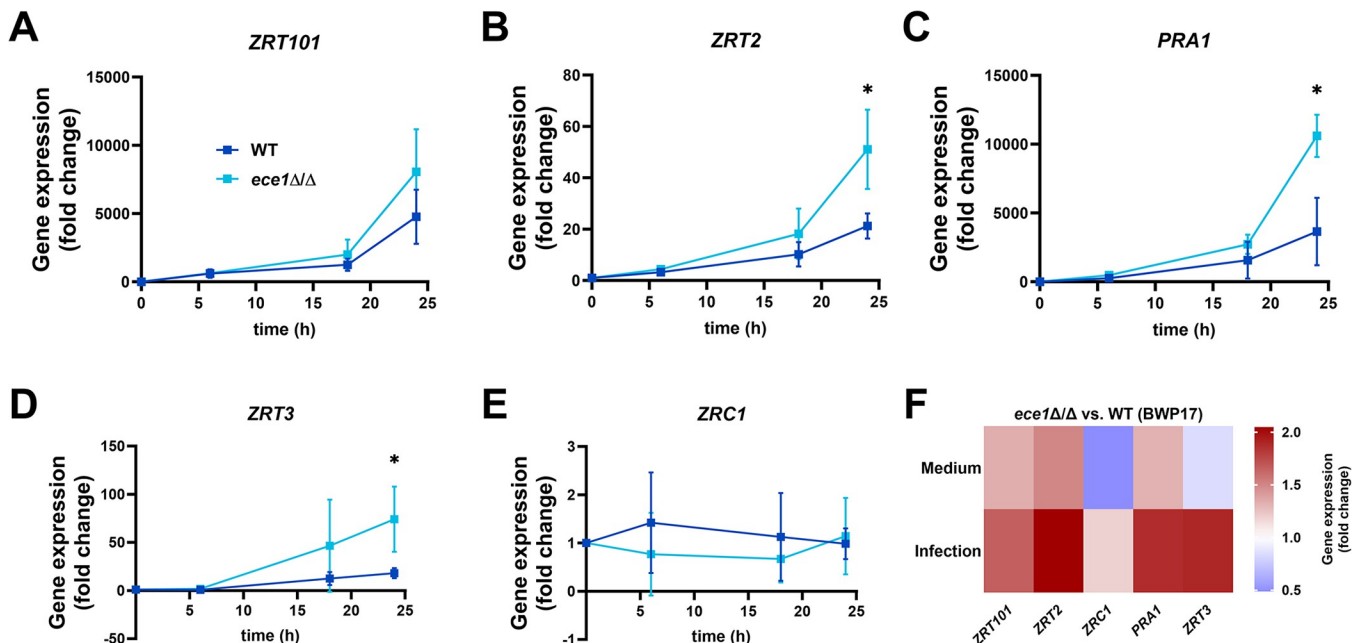

**Fig 3. Loss of *ECE1* increases the transcriptional zinc starvation response during infection of IECs.** Fold change in gene expression in *C. albicans* WT (BWP17) and *ece1*Δ/Δ strains during co-incubation with C2BBe1 cells for (A) *ZRT101*, (B) *ZRT2*, (C) *PRA1*, (D) *ZRT3*, and (E) *ZRC1*. Fold changes are calculated as the normalized gene expression at each time point compared to the respective 0 h control samples. (F) Fold change in normalized gene expression of the *ece1*Δ/Δ strain compared to WT(BWP17) at 24 h in medium only or during infection of IECs. Gene expression was analyzed by q-RT-PCR and is normalized to *ACT1* as a housekeeping gene. All values are shown as the mean with standard deviation. All the ratio data from q-RT-PCR experiments were log-transformed before performing a two-tailed, paired t-test. Statistical significance: *, $P \leq 0.05$.

efficient zinc acquisition by *C. albicans* appears necessary to fully damage IECs, likely by fostering its growth and filamentation.

We have shown that *C. albicans* experiences more intense late-stage zinc starvation in the absence of host cells–at time points at which damage of the IECs takes place in infected samples (Fig 1C) [11]. We therefore hypothesized that *C. albicans* acquires zinc from host cells during infection of the intestinal epithelium and that host-cell damage enables this zinc acquisition. The fungal peptide toxin candidalysin, encoded by the *ECE1* gene, is the main damaging factor of *C. albicans* [11,14]. To further investigate the interplay between host-cell damage and zinc acquisition and the involvement of candidalysin in this process, we compared the transcriptional zinc starvation response of wild-type (WT, BWP17) *C. albicans* to an *ece1*Δ/Δ strain. The strain lacking *ECE1* showed significantly higher transcript levels of *ZRT2*, *PRA1*, and *ZRT3* after 24 h of infection of IECs, consistent with more severe zinc starvation (Fig 3B, 3C, 3D and 3F). A similar pattern was seen for *ZRT101* mRNA levels, though the difference was not statistically significant (Fig 3A and 3F). When cultured in the absence of host cells, the WT (BWP17) and *ece1*Δ/Δ strains showed no significant differences in transcript levels of zinc-related genes (Fig 3F and S3A–S3E Fig). The addition of exogenous zinc during co-incubation with IECs did not affect the host-cell damage, fungal translocation, or fungal mass of the WT(BWP17) or *ece1*Δ/Δ strains during infection of IECs (S4A–S4C Fig). The differences in the transcriptional zinc starvation response between the WT (BWP17) and *ece1*Δ/Δ strains after 24 h were reduced with addition of zinc, though there was still a trend towards increased transcript levels in the *ece1*Δ/Δ strain (S4D Fig). These data indicate, that *ECE1* is necessary for zinc acquisition during interaction with IECs.

## IEC response to *C. albicans* infection by NFκB, MAPK, and TNF signaling and damage-dependent c-Fos/IL-8 induction

Similar to *C. albicans*, IECs showed a time-dependent transcriptional response with patterns reflecting the different stages of infection (Figs 1A and 5A). The transcriptomes of infected host cells differed from those cultured without *C. albicans* at all time points, showing a clear infection-specific response (Fig 4A). The time-dependent response of IECs presented as a drastic increase in the number of infection-specific DEGs after 12 and 24 h–time points at which *C. albicans* hyphae damage the epithelium and translocate (S1 Table) [11]. KEGG (S2 Table and Fig 4B) enrichment analysis was performed on the infection-specific IEC DEGs. Similar to *C. albicans*, there were few significant infection-specific pathways at the 45 min time point (over-representation analysis, adjusted p ≤ 0.05; Fig 4B). Components of the JNK and p38 MAPK pathway were induced upon early infection, including the c-Jun N-terminal kinase (JNK) gene *MAPK10* as well as the transcription factor AP-1 components *FOS*, *JUN* and *JUNB*. *FOS* transcript levels remained high during infection at all later time points (S1 and S2 Tables).

Soon after initial hypha formation at 3 h, we observed an infection-specific increase in mRNA levels of epithelial genes involved in oxidative phosphorylation (Fig 4B). Specifically, transcript levels were higher for mitochondrial genes (*ND1*, *2*, *4*, *5*, *6*; *ND4L*; *COX1*, *2*, *3*; *ATP6*, *8*; *CYB*), which has previously been observed in vaginal epithelial cells [19] (S1 and S2 Tables). After 6 h when invasion of host cells occurs, and until the final time point at 24 h, genes involved in innate immune responses were significantly overrepresented in the infection-specific DEGs. MAPK, TNF, NFκB, and IL-17 signaling pathways were overrepresented due to genes with increasing transcript levels during IEC infection (Fig 4B). Genes involved in the classical MAP kinase pathway (including growth factor genes *EFNA1*, *EREG*, *AREG*, *TGFA*; calcium voltage-gated channel *CACNA* genes; receptor tyrosine kinase genes *NGFR*, *EPHA2*, *FGRF1*; protein kinase C gene *PRKCB*; protein tyrosine phosphatase and mitogen-activated protein kinase phosphatase genes *PTPN7*, *DUSP2*, *4*, *5*, *6*, *10*; NFKB2) had increased transcript levels upon *C. albicans* infection (S1 and S2 Tables). Genes encoding chemokines (*CXCL1*, *2*, *3*, *8*, *17*), inflammatory cytokines (*IL1B*, *IL11*, *IL12A*, *IL27*, *IL32 CCL20*), as well as genes for intracellular signaling proteins (*GADD45A*, *B*; *FOS*; *FOSL1*; *FOSB*; *JUN*; *A20*) showed increased transcript levels upon *C. albicans* infection (S1 and S2 Tables). The late-stage induction of innate immune signaling pathways is consistent with data previously published for IECs [18].

The majority of infection-specific host DEGs only appear after fungal invasion and resulting host-cell damage. In comparison, vaginal epithelial cells show a uniform early response to various fungal species, but the response diverges at the later time points and is connected to damage potential [19]. We sought to determine whether the IEC DEGs at later time points reflect a general response to fungi or rather a specific response to *C. albicans* filamentation and damage. To determine which of the host responses identified were filamentation- or damage-specific, we extended our transcriptome analysis to compare IECs infected with *C. albicans* WT, non-damaging (*ece1Δ/Δ*), and non-filamentous and non-damaging (*efg1ΔΔ/cph1ΔΔ*) strains. The two mutant strains used (*ece1Δ/Δ* and *efg1ΔΔ/cph1ΔΔ*) were constructed in different genetic backgrounds, therefore the gene expression data was always compared to the corresponding WT strains (BWP17 and SC5314, respectively) to minimize the effect of strain-specific, genetic differences on the host response [30]. The set of genes that were only differentially expressed in response to the *efg1ΔΔ/cph1ΔΔ* strain was then categorized as a filamentation-specific response. Genes that were differentially expressed in response to both the *efg1ΔΔ/cph1ΔΔ* and *ece1Δ/Δ* strains, were considered to constitute a damage-specific response. The filamentation-specific response of IECs to *C. albicans* did not consist of many genes, though it

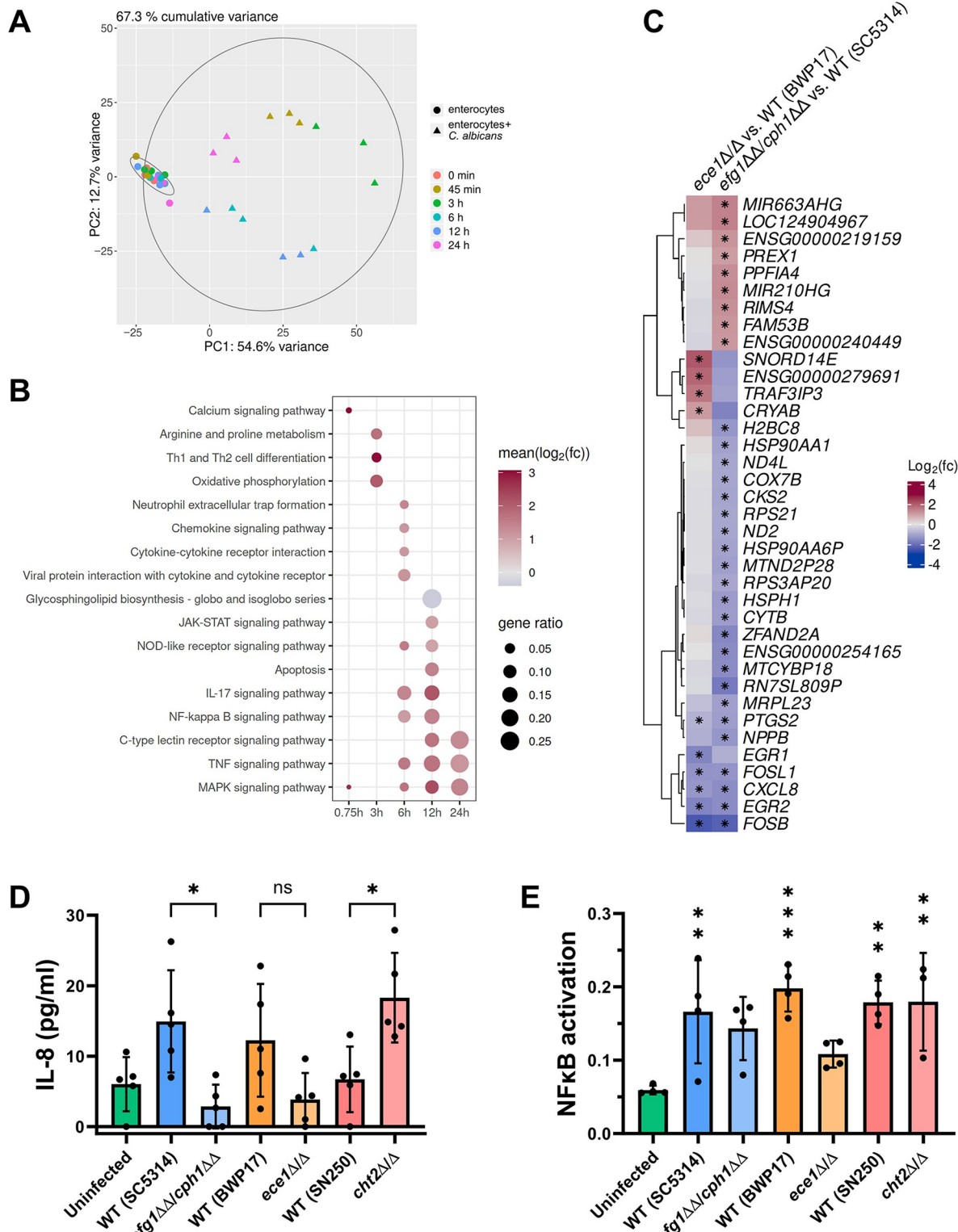

**Fig 4. The transcriptional response of IECs to *C. albicans* infection is largely damage- and filamentation-independent.** (A) Principal component analysis for IEC RNA sequencing data. Samples from IECs (enterocytes) cultured in medium only cluster together across all time points and separately from those co-incubated with *C. albicans*. (B) KEGG categories significantly overrepresented among infection-specific DEGs in IECs. Gene ratio represents the proportion of genes within each category that were significantly differentially expressed during infection compared to IECs only. The color scale represents the mean Log$_2$(fold-change) of significant DEGs within each category.

Red indicates that most DEGs showed increased expression during infection, blue indicates most DEGs showed decreased expression, and grey indicates a mix of DEGs with increased and decreased expression within the category. Dots are shown only for significantly enriched categories (overrepresentation analysis, Benjamin-Hochberg adjusted p-value ≤ 0.05, S2 Table). (C) Significantly differentially expressed genes between *ece1Δ/Δ* and *efg1ΔΔ/cph1ΔΔ C. albicans* strains compared to their respective WT strains during infection of IECs. The color scale represents the Log$_2$(fold-change) for each gene during infection in the mutant strain compared to the respective WT strain, with red representing higher expression in the mutant strain and blue representing lower expression in the mutant strain. An asterisk indicates the genes with a statistically significant difference (DESeq2 p < 0.05). The mean MRN values are given in S1 Table. (D) Release of IL-8 by C2BBe1 cells infected with *C. albicans* strains possessing varying damage potentials. The non-damaging *efg1ΔΔ/cph1ΔΔ* and *ece1Δ/Δ* strains induced less IL-8 release, though only significantly for the *efg1ΔΔ/cph1ΔΔ* strains. Conversely, the high-damaging *cht2Δ/Δ* strains induced significantly more IL-8 release. (E) NFκB activation as measured by the DNA-binding activity of the p65 transcription factor. Infection with all *C. albicans* strains increased the DNA-binding activity of p65 compared to the uninfected C2BBe1 cells, though not to a statistically significant degree for the *efg1ΔΔ/cph1ΔΔ* and *ece1Δ/Δ* strains. All values are shown as the mean with standard deviation. The IL-8 (D) and NFκB activation (E) data were compared using a one-way ANOVA with a post-hoc Šidák's multiple comparisons test. Statistical significance: *, $P \leq 0.05$; **, $P \leq 0.01$; ***, $P \leq 0.001$.

included mitochondria-associated genes (*COX7B, ND2, CYTB, ND4L*) and some encoding heat-shock proteins (*HSP90AA1* and *HSP90AA6P*), all with lower transcript levels in the non-filamentous mutant as compared to the WT (SC5314) (Fig 4C and S5 Fig).

The damage-specific response of IECs comprises even fewer genes, which in most cases showed lower transcript levels in IECs infected with the *ece1Δ/Δ* and *efg1ΔΔ/cph1ΔΔ* mutants than with the respective WT strains. According to these data, IECs respond to *C. albicans*-mediated damage *via* upregulation of *EGR1* and *EGR2*, the FOS genes *FOSL1* and *FOSB* as well as the IL-8-encoding gene *CXCL8* (Fig 4C and S5 Fig). c-Fos is a major component of the oral epithelial cell (OEC) response to *C. albicans* filamentation and damage [13, 14]. Similarly, EGR1 induction as well as IL-8 secretion are known responses to *C. albicans* infection in OECs [31].

To test how conserved the response of IECs and OECs to *C. albicans* and fungus-mediated damage are, we performed western blotting for the OEC response components EphA2 and EGFR (receptors), the transcription factor c-Fos, the phosphatase MKP1 and the kinases Akt and p38 [13,16,17,32]. We observed an infection-specific increase in phosphorylation of EphA2 and Akt, as well as an infection-specific increase in c-Fos protein levels. c-Fos signals were consistently reduced in the *ece1Δ/Δ* mutant. While there was also decreased phosphorylation of EphA2 for the *ece1Δ/Δ* mutant compared to the WT, this was not consistent for all replicates (S6A and S6B Fig). Thus, we see a partial conservation in the damage response during *C. albicans* infection of IECs compared to OECs, with *ECE1*-dependent induction of c-Fos and potentially phosphorylation of EphA2, but no EGFR or MKP1 phosphorylation and no change in the phosphorylation of p38 or Akt in the absence of *ECE1* compared to the WT [14–16,33,34].

To verify that production of IL-8 in IECs is filamentation- and damage-dependent, we measured IL-8 secretion in response to *C. albicans* infections. We compared the WT with the non-filamenting *efg1ΔΔ/cph1ΔΔ* strain, the non-damaging *ece1Δ/Δ* strain, and a previously identified strain (*cht2Δ/Δ*) with increased damage of IECs [11] (Fig 4D). Indeed, the *efg1ΔΔ/cph1ΔΔ* and *ece1Δ/Δ* mutants elicited less IL-8 secretion while *cht2Δ/Δ* induced more IL-8 secretion than their respective WT strains. Overall these data show that while the response of IECs to *C. albicans* infection is largely independent of hypha formation and fungal-mediated damage, there is a damage-specific aspect *via* c-Fos induction and IL-8 secretion.

## Intestinal epithelial cells limit fungal damage and translocation *via* NFκB activation

Most of the pathways overrepresented during infection with WT *C. albicans* were also found during infection with the non-damaging and non-filamentous strains (S2 Table), again suggesting that the IEC transcriptional response to *C. albicans* is largely independent of fungal filamentation or damage. Among other signaling pathways that were triggered by the initiation

of fungal-mediated damage (Fig 4B), DEGs were enriched in NFκB signaling after infection with all four *C. albicans* strains (S2 Table). Previous research has shown that *C. albicans*-induced NFκB activation in IECs limits the damage potential during infection [18]. We confirmed the *C. albicans*-induced NFκB activation in IECs and additionally showed that infection with *C. albicans* strains with high or low damage potential and even with a non-filamentous mutant elicit similar activation of NFκB by IECs (Fig 4E). This suggests that NFκB activation is a general epithelial response to the presence of *C. albicans* (Fig 4E).

To determine whether the previously described NFκB-dependent limitation of IEC damage is predominantly host- or fungal-driven, IECs were infected with the non-filamentous *efg1ΔΔ/cph1ΔΔ* and non-damaging *ece1Δ/Δ* strains in the presence and absence of the potent NFκB activation inhibitor quinazoline (QNZ) [18, 35]. In agreement with previous data, both tested WT strains elicited significantly more host-cell damage when NFκB activation was blocked (Fig 5A) [18]. In contrast, there was no statistically significant increase in damage caused by the strains lacking either *ECE1* or both *EFG1* and *CPH1* (Fig 5A). Treatment of IECs with a DMSO vehicle control had no significant effect on either the damage potential or fungal translocation for the WT (BWP17) and *ece1Δ/Δ* strains (S7A and S7B Fig).

As host-cell damage of IECs is tightly linked to fungal translocation, the translocation rates of *C. albicans* in the presence of QNZ were also measured [11]. As expected from our host-cell damage data, blocking NFκB activation significantly increased the number of translocated fungal CFUs for both WT strains compared to the untreated controls (Fig 5B). The non-filamentous *efg1ΔΔ/cph1ΔΔ* strain showed no increase in its translocation rate, which was expected given that this strain cannot form hyphae or damage the host cells (Fig 5A and 5B) [11,36]. Surprisingly, despite showing no significant increase in host-cell damage upon QNZ-treatment [<50% of untreated WT (BWP17)], the *ece1Δ/Δ* strain showed a significant increase in translocation comparable to that of the WT (BWP17) under normal infection conditions (Fig 5B). This increased translocation in the absence of increased host-cell damage was confirmed for the *ece1Δ/Δ* strain with the use of another NFκB inhibitor with a different mode of action that directly inhibits the DNA-binding activity of the p65 subunit [37,38] (S8A and S8B Fig). These data together indicate that NFκB can limit *C. albicans* translocation, at least in part, independently of host-cell damage.

We have previously shown that *C. albicans* translocation across IECs occurs mainly *via* a transcellular route and is associated with filamentation- and candidalysin-dependent necrotic host-cell damage [11]. The increased translocation of the *ece1Δ/Δ* strain without any increased host-cell damage upon NFκB inhibition suggests that NFκB activation likely limits a non-damaging route for translocation (Fig 5B). To test this hypothesis, the transepithelial electrical resistance (TEER) of IECs was measured with and without QNZ treatment as a metric for barrier integrity. After 24 h, IECs infected with both WT strains showed decreased barrier integrity upon QNZ treatment, though not to a statistically significant degree (Fig 5C) [WT (SC5314), *P* = 0.2466; WT (BWP17), *P* = 0.0824]. These results match the increased host-cell damage and translocation data for both strains. There was also a significant decrease in the barrier integrity for IECs infected with both, the *efg1ΔΔ/cph1ΔΔ* and *ece1Δ/Δ* strains upon QNZ treatment, again pointing to a host damage-independent effect of NFκB activation on translocation (Fig 5C). This was also confirmed for the WT and *ece1Δ/Δ* strains using the second NFκB inhibitor (S8A–S8C Fig).

Non-damaging routes of fungal translocation that still reduce barrier integrity can involve induction of host-cell apoptosis or the use of paracellular routes with degradation of cell-cell connections such as tight junctions or adherens junctions [11]. As the NFκB signaling pathway can also induce the expression of anti-apoptotic genes and we observed an overrepresentation of infection-specific DEGs for IECs involved in apoptosis at 12 h when fungal-mediated

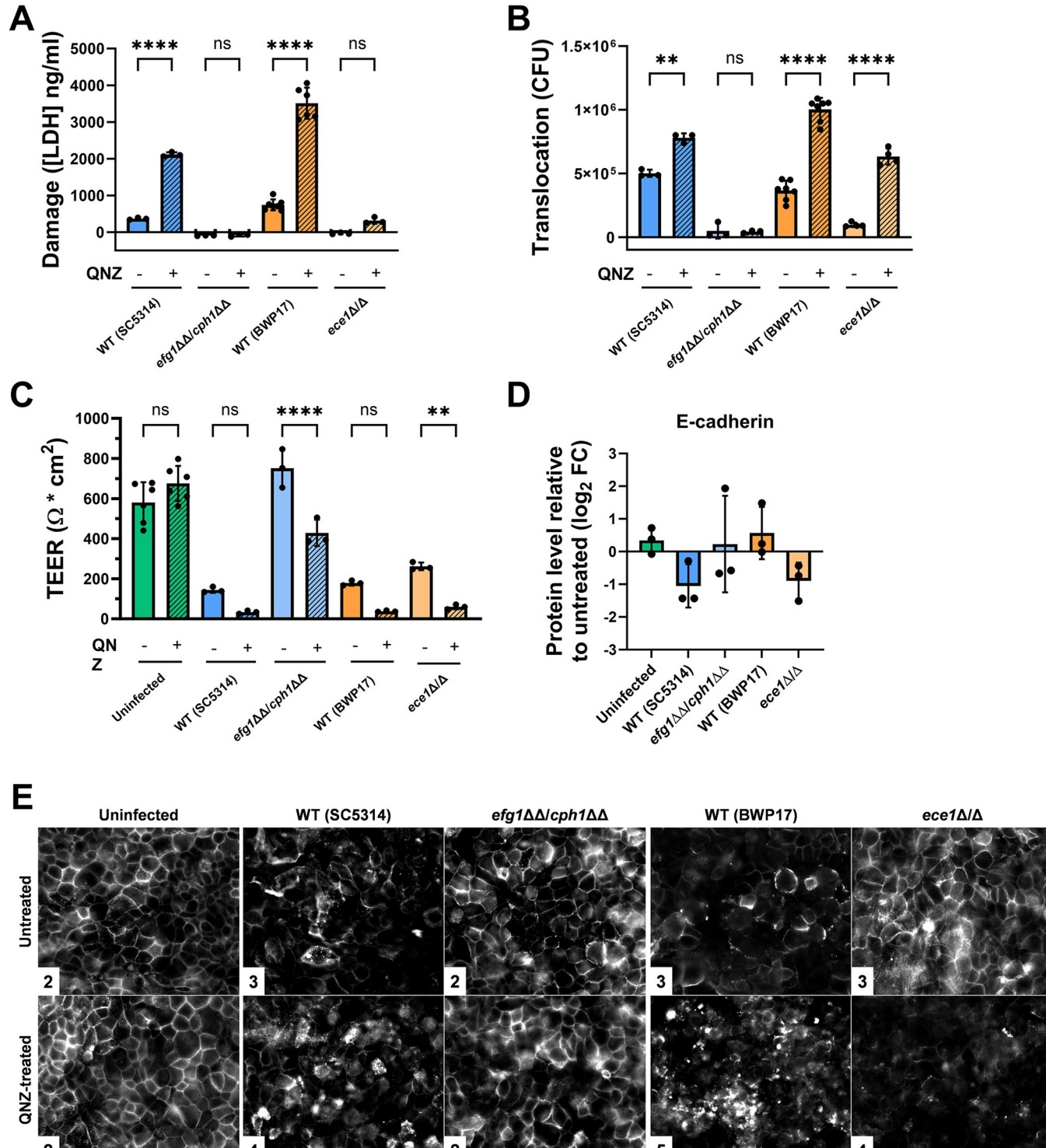

**Fig 5. NFκB activation limits *ECE1*-dependent damage and paracellular translocation.** (A) Inhibition of NFκB activation using the high affinity NFκB activation inhibitor quinazoline (QNZ) increases the damage potential of *C. albicans* wildtype, but not for the non-filamentous *efg1ΔΔ/cph1ΔΔ* strain or the non-damaging *ece1Δ/Δ* strain towards C2BBe1 cells after 24 h. LDH release was adjusted by subtracting the release from uninfected and untreated host cells. (B) Inhibition of NFκB activation increases fungal translocation of *C. albicans* after 24 h across intestinal epithelial cells independent of damage potential. (C) Blocking NFκB activation significantly increases the breakdown of barrier integrity after 24 h of the intestinal epithelial cell layer for *efg1ΔΔ/cph1ΔΔ* and *ece1Δ/Δ*, with a similar but not significant effect on both WT strains. (D) E-cadherin protein levels normalized to GAPDH and presented relative to levels in untreated C2BBe1 cells. QNZ treatment further increased degradation of E-cadherin during infection with WT (SC5314) and *ece1Δ/Δ*. (E) Fluorescent labeling

of E-cadherin during *C. albicans* infection with and without QNZ treatment (scale bar = 50 μm). Inhibition of NFκB activation decreased the organization of E-cadherin at IEC borders upon infection with the *ece1Δ/Δ* and both WT strains. Representative pictures for each strain and treatment condition are shown from 3 biological replicates with median score values in the inset for each strain and condition. All values are shown as the mean with standard deviation. Host-cell damage (A), fungal translocation (B), and barrier integrity (C) data were compared using a one-way ANOVA. Statistical significance for host-cell damage (A) and fungal translocation (B) data was determined with a post-hoc Tukey's multiple comparisons test, while significance for the barrier integrity (C) data was determined using a post-hoc Šidák's multiple comparisons test. Statistical significance: **, $P \leq 0.01$; ****, $P \leq 0.0001$.

damage begins (Fig 4B) [39]. We therefore tested whether blocking NFκB activation during infection with *C. albicans* increases apoptosis or other types of programmed cell death in IECs. IECs were left untreated or treated with QNZ, a pan-caspase inhibitor (Z-VAD), or both, and were then infected with different *C. albicans* strains. Treatment with Z-VAD alone had no significant effects on host cells damage, loss of barrier integrity, or fungal translocation (S9A–S9C Fig). Similarly for IECs with blocked NFκB activation, addition of Z-VAD led to no significant differences with any of the strains tested (S8A–S8C Fig). Together, these data suggest that the ability of IECs to limit cellular damage and translocation through NFκB activation is dependent on the filamentation and damage potential of *C. albicans* itself, and does not rely on increased apoptosis or other forms of programmed cell death.

To finally test whether the decreased barrier integrity during infection with the *ece1Δ/Δ* strain was associated with increased degradation of cell-cell junction proteins consistent with increased paracellular translocation, epithelial barriers were stained for E-cadherin 24 h after infection with and without QNZ treatment. Micrographs from three biological replicates were then scored by a blinded observer for the consistency of the cell-cell borders and overall organization of the E-cadherin staining. Non-inhibitor treated uninfected and *efg1ΔΔ/cph1ΔΔ*-infected IECs showed a uniform staining of E-cadherin, indicating an intact host cell layer (Fig 5C and 5E). In contrast, untreated cells infected with either WT strain or the *ece1Δ/Δ* strain showed a weaker staining (Fig 5E). Inhibitor treatment further decreased the E-cadherin signal in IECs infected with either of the two WT strains, as expected from the inhibitor-dependent increase in damage (Fig 5A). Importantly, the *ece1Δ/Δ* strain showed less intact host cells and decreased fluorescence upon NFκB activation inhibition, while the host cell layer was largely unchanged for the uninfected and *efg1ΔΔ/cph1ΔΔ*-infected IECs when treated with QNZ (Fig 5E). These data indicate that NFκB limits paracellular translocation by stabilizing cell-cell junction proteins like E-cadherin. They also suggest that, upon NFκB inhibition, the physical presence of hyphae can break down intercellular barrier function even in the absence of toxin-mediated damage.

To further investigate the effects of NFκB inhibition on cell-cell junction proteins at a molecular level, we measured the junction proteins E-cadherin and claudin-1 by western blotting for infected IECs with and without QNZ treatment (S10A Fig). Western blotting showed that the total protein levels of E-cadherin and claudin-1 (normalized to GAPDH protein levels) in *C. albicans* infected IECs were largely unaffected by NFκB inhibition (Figs 5D and S10B Fig). There was no consistent difference in E-cadherin levels upon inhibitor treatment during infection with the WT (BWP17) or *efg1ΔΔ/cph1ΔΔ* strains. However, relative E-cadherin levels upon treatment were reduced for IECs infected with the WT (SC5314) and *ece1Δ/Δ* strains, though not to a statistically significant degree (Fig 5D). The same trend was true for the *ece1Δ/Δ* and *efg1ΔΔ/cph1ΔΔ* strains for claudin-1 (S10B Fig). Though the decrease in both E-cadherin and claudin-1 for the *ece1Δ/Δ* strain was not significant, it did correlate with the decrease in barrier integrity (Fig 5C) and is also consistent with the organization of the cell-cell junction proteins (Fig 5E). In summary, these data show that NFκB activation limits *ECE1*-mediated host-cell damage and independently limits fungal translocation. It does do by maintaining cell-cell junctions and the epithelial barrier integrity in the absence of direct fungal-mediated host-cell damage.

## Discussion

In this study, we explored fungal and host factors that contribute to *C. albicans* translocation across intestinal epithelial cell layers, a process which precedes systemic infections. Dual-species RNA sequencing of *C. albicans* and human cells revealed the contribution of fungal zinc acquisition to host-cell damage. We confirmed a partially conserved damage-specific response of IECs to *C. albicans* infection and dissected the role of NFκB activation in limiting fungal-mediated damage and translocation.

The *C. albicans* infection-specific transcriptional response was largely time dependent, with most DEGs appearing at the later stages of IEC infection. This was expected as the time points chosen are associated with different stages of epithelial infection: from adhesion, hypha formation, and invasion to host-cell damage and fungal translocation [11]. There was no enrichment of known filamentation- or virulence-related terms at time points later than 45 min. This is likely due to the fact that the expression of many *C. albicans* virulence factors is associated with the yeast-to-hypha transition [40,41]. In our experimental setup, there were strong triggers for filamentation in both the infected and medium-only control samples, and hyphae were present at all experimental time points from 3 to 24 h. Transcriptional changes due to the switch from YPD to DMEM medium may account for some similarities in the transcriptional profiles between the infected and control samples at the earlier timepoints. However, the similar pre-culture conditions have been shown previously to play no major role in the gene expression pattern of *C. albicans* already after 30 min [42]. At around 6 h when epithelial invasion occurs, we however observed an infection-specific increase in transcript levels for genes involved in ergosterol biosynthesis. This may be due to subtle physiological changes that support invasion and damage of IECs, as ergosterol biosynthesis is linked to filamentation in *C. albicans* [43]. This is followed by infection-specific changes in the transcript levels of cell adhesion genes at 12 h. These data suggest that the interaction with IECs induces *C. albicans* hyphae that are physiologically distinct to hyphae grown without any contact to host cells. Taken together, our data along with that of previous studies indicate that while many different environmental conditions trigger morphologically similar hyphae in *C. albicans*, their precise molecular compositions depend on the growth conditions in which they form [44,45].

The majority of infection-specific changes in the fungal transcriptome were metabolic in nature. Our data show faster glucose consumption during IEC infection, indicated by decreased transcript levels of glycolysis genes after 12 h. This was unsurprising as both fungus and host compete for glucose during infection. In line with this, previous data showed glucose levels were lower during IEC infection by *C. albicans* after 12 h compared to the fungus grown in medium alone [46]. *C. albicans* further adjusted its metabolism during infection by altering the transcript levels of genes involved in sugar transport and in amino acid metabolism. After 24 h, when a large portion of the epithelial cells was damaged, there were further fungal metabolic adaptations, especially increased mRNA levels of genes involved in glycolysis and galactose metabolism. As transcription of galactose-metabolic *GAL* genes is induced in the absence of glucose [47], this may suggest that *C. albicans* gains access to new non-glucose carbon sources which are released due to extensive host-cell damage.

It has been hypothesized that *ECE1*/candidalysin-mediated cytolysis may be necessary for nutrient acquisition from host cells during fungal infection [48,49]. Our study provides evidence in support of this hypothesis by showing an *ECE1*-dependent alleviation of zinc starvation during epithelial invasion. This suggests that invasion may contribute to zinc acquisition from the host. Zinc is a vital micronutrient for both pathogen and host alike. The availability of zinc for *C. albicans* within the host is limited during systemic infection and oropharyngeal candidiasis due to nutritional immunity [50–53]. Depending on the host niche, *C. albicans* can

also encounter high and potentially toxic levels of zinc, such as following phagocytosis by immune cells. While *C. albicans* is limited for zinc during systemic infection, the fungus does not experience severe zinc starvation during infection of IECs [50,51].

Our observations indicate that *C. albicans* faces a nutritionally complex environment during IEC infection. In general, the transcript levels of the zinc uptake genes *ZRT101*, *ZRT2*, and *PRA1* were higher in medium alone than during IEC co-culture. This shows that *C. albicans* experiences less zinc starvation in presence of IECs and suggests that it can access zinc from the host. Deletion of either component of the Pra1/Zrt101 zincophore system significantly reduced the capacity of *C. albicans* to damage IECs. The *zrt2Δ/Δ* mutant was unaffected, and the *zrt101ΔΔ/zrt2ΔΔ*, which is defective in both known zinc uptake pathways, caused no damage. This suggests that *C. albicans* primarily utilizes the Pra1/Zrt101 system to acquire zinc from IECs, and that the Zrt2 importer can play an important role in the absence of a functional zincophore. In line with this, zinc supplementation of the media fully restored growth and damage potential of zinc uptake-defective strains.

In the absence of exogenous zinc supplementation, the only zinc source for *C. albicans* is the host epithelium itself. In line with this, preventing host cell cytolysis by *ECE1* deletion resulted in a greatly amplified zinc starvation response, which could again be partially alleviated by exogenous zinc supplementation. These results also fit well with clinical manifestations of candidemia. While the human gastrointestinal tract normally contains high levels of zinc in various complexed forms, zinc deficiency is recognized as a significant risk factor for patients in the pediatric intensive care unit, and oral zinc supplementation is recommended as a prophylaxis [54–56]. Imbalanced zinc homeostasis within the intestine is also associated with microbial dysbiosis and other intestinal diseases that impair the barrier function of the intestinal epithelium, such as inflammatory bowel disease, irritable bowel syndrome and colorectal cancer [57]. In our *in vitro* models of IEC infection, loss of *ECE1* did not affect fungal growth *per se* and thus addition of exogenous zinc was not sufficient to restore normal host-cell damage or fungal translocation despite the partially alleviated starvation response. This could be due to increased sensitivity of our transcriptional analysis in our experimental model or compensatory activities in zinc mobilization of fungal cells that are sufficient to allow for normal growth. Additionally, most fungal material in our models grows atop the epithelial cells and not invasively. Thus, more complex model systems for prolonged intestinal epithelial invasion would be necessary to determine the contribution of Ece1 to nutrient acquisition from the host, especially the physiological relevance of *ECE1*-mediated zinc acquisition.

Interestingly, *zrc1Δ/Δ* exhibited the most severe defect in invasive growth and IEC translocation; it also caused much less damage than the WT, which was not reversed upon zinc supplementation. The primary function of Zrc1 is to detoxify fungal cytoplasmic zinc *via* compartmentalization within vesicular-like structures called zincosomes [28]. This, together with the reduced expression of zinc uptake genes by *C. albicans* on IECs suggests that the fungus requires either Zrt101/Pra1 or Zrt2 to capture zinc from the host, but that once internalized, the zinc detoxification system is crucial for invasion, host-cell damage, and fungal translocation.

The epithelial infection-specific transcriptional response was also dependent on the time point, though to a lesser degree than that of *C. albicans*. We observed increased transcript levels of mitochondria-associated genes. As this has been shown previously in vaginal epithelial cells as well, it indicates a potentially conserved early epithelial response to fungal infections *via* mitochondrial signaling [19]. The infection-specific host response at later time points was characterized by MAPK, TNF, and NFκB signaling consistent with earlier findings [18]. Using our data, we were able to dissect the conserved specificity of this response in a time-resolved manner and to identify additional aspects of the IEC response and its influence on the

interaction with *C. albicans*. At 12 h when host-cell damage and fungal translocation begin, our data show that there is a limited damage- and filamentation-specific transcriptional response to *C. albicans* infection. We could, however, show that IECs possess a damage-specific response to *C. albicans* partially conserved with other epithelial cell types [13–17]. Specifically, IECs show increased expression of *FOSB*, *FOSL1*, and *CXCL8* in the presence of filamentation and candidalysin. Furthermore, IECs show induction of c-Fos and increased phosphorylation of EphA2 and Akt, though this was largely independent of host-cell damage. This indicates that IECs possess a conserved, albeit reduced, damage-specific response to *C. albicans* infection compared to OECs. This is further exemplified by IL-8 secretion of IECs which correlates with the damage potential of the infecting *C. albicans* strain, but is much lower overall as compared to oral and vaginal epithelial cells [19,31].

Our data show that the pathways induced in IECs are largely independent of *C. albicans* host-cell damage and filamentation. Activation of the NFκB signaling pathway in IECs was common for all *C. albicans* strains tested, and it has already been shown to limit the damage potential of *C. albicans* [18]. We show that this effect on host-cell damage is dependent on the *ECE1* gene, as inhibiting NFκB activation had no significant effect on the damage potential of the *ece1Δ/Δ* strain. Blocking NFκB activation also increased fungal translocation, though this was independent of *ECE1*, suggesting that NFκB activation limits fungal-mediated damage and fungal translocation *via* separate mechanisms. We first hypothesized that NFκB activation could limit fungal-mediated damage by inducing expression of anti-apoptosis genes and preventing programmed cell death [39]. However, a pan-caspase inhibitor showed no effect on the damage potential of *C. albicans* and did not counteract the increased host-cell damage when NFκB activation was inhibited. These data suggest that NFκB activation upon *C. albicans* infection does not prevent apoptosis or other programmed cell death pathways. The specific mechanism by which NFκB signaling limits fungal-mediated damage remains undetermined, but may involve recently described membrane repair mechanisms of epithelial cells [58,59].

We found increased translocation upon inhibition of NFκB activation to be associated with increased breakdown of the epithelial barrier integrity, independent of fungal host-cell damage potential. This contrasts with an earlier study in which NFκB inhibition did not influence IEC barrier breakdown by *C. albicans* [18]. This discrepancy could be attributed to the different experimental conditions between the previous study and ours. While in our study, there were low remaining levels of transepithelial electrical resistance detected after 24 h of *C. albicans* infection in the absence of an NFκB inhibitor, no transepithelial electrical resistance measurement could be detected at this time point in the previous study [18]. This could be due to different differentiation times for the IECs, cell culture media, pore sizes of the transwell membranes, or the MOIs used for infection. Our results rather suggest that NFκB activation may limit fungal translocation *via* the paracellular route [6].

We found the protective function of NFκB signaling on the barrier integrity during *C. albicans* infection to be associated with reduced degradation of cell-cell junction proteins. Under normal experimental conditions, *C. albicans* strains that filament and produce sufficient amounts of candidalysin are likely able to overcome this and translocate following necrotic host-cell death [11]. Strains with low-damage potential, like *ece1Δ/Δ*, are unable to take this route and are thus significantly diminished in their ability to translocate. In this study, we used immortalized intestinal cell lines as a model to dissect the specific cellular mechanisms behind fungal translocation. Future experiments with more complex *in vitro* model systems that incorporate more physiological factors, like organ-on-chip or intestinal organoids [60–62], will help to understand these interactions better. Nevertheless, our findings provide mechanistic insights into how conditions or treatments that alter activation of NFκB could put patients at risk for systemic candidiasis, such as those undergoing treatment with steroids [1,2,39,63].

Host-cell damage and efficient translocation across the intestinal epithelium by *C. albicans* are dependent on candidalysin, though both processes also require other fungal factors [11,12]. Our data connect zinc acquisition from intestinal epithelial cells to fungal growth and host-cell damage during infection, and reveal a limited damage-dependent response of intestinal epithelial cells to *C. albicans*. Activation of NFκB signaling in intestinal epithelial cells increased the barrier integrity during infection, which limited fungal translocation especially for *C. albicans* strains with low damage potential. This suggests that host defense mechanisms at the intestinal epithelium limit paracellular translocation, forcing *C. albicans* towards the damage-mediated transcellular path to overcome the intestinal barrier.

## Materials and methods

### *Candida albicans* strains and growth conditions

*C. albicans* strains used in this study are shown in Table 1. The WT strains SC5314 and BWP17+CIp30 are referred to as WT (SC5314) and WT (BWP17), respectively. Strains were routinely cultivated on/in YPD agar/broth (1% yeast extract, 2% peptone, 2% D-glucose with or without 1.5% agar) at 30°C. Overnight (O/N) cultures were cultured for 16 h in YPD broth at 30°C with shaking at 180 rpm unless otherwise specified. Cells were then washed twice with phosphate-buffered saline (PBS) and the cell number was adjusted.

For experiments with prior zinc starvation, *C. albicans* strains were grown for 24 h in YPD liquid medium supplemented with 1 mM $ZnSO_4$ at 30°C with shaking at 180 rpm. The $OD_{600}$ was then adjusted to 0.5 in SD liquid medium supplemented with 1 mM $ZnSO_4$ and incubated at 30°C for 24 h with shaking at 180 rpm. The cells were then washed twice with 1 mM EDTA, washed twice with water, and then diluted to an $OD_{600}$ of 0.5 in Zn-free SD liquid medium in acid-washed, plastic flasks. The fungal strains were then incubated for another 24 h at 30°C with shaking at 180 rpm.

### Culture of intestinal cells

The intestinal epithelial Caco-2 brush border expressing 1 cell line (C2BBe1; ATCC, CRL2102) [68] and the human intestinal goblet cell line (HT29-MTX; ATCC, HTB-38; CLS, Lot No. 13B021) were routinely cultivated in Dulbecco's Modified Eagle's Medium (DMEM) (Gibco, Thermo Fisher Scientific) supplemented with 10% fetal bovine serum (FBS) (Bio&Sell), 10μg/

Table 1. *C. albicans* strains used in this study.

| *C. albicans* strains | Parental strain | Relevant genotype | Source |
|---|---|---|---|
| SC5314 | | wild type, clinical isolate | [64] |
| BWP17+CIp30 | | *ura3::λimm434/ura3::λimm434 his1::hisG/his1::hisG arg4::hisG/arg4:: hisG* +CIp30 | [65] |
| SN250 | | Noble wild type (leu2::CdHIS1/leu2::CmLEU2) | [66] |
| *efg1ΔΔ/cph1ΔΔ* | SC5314 | *efg1::FRT/efg1::FRT cph1::FRT/cph1::FRT* | [67] |
| *ece1Δ/Δ* | BWP17 | *ece1::HIS1/ece1::ARG4 RPS1/rps1::URA3* | [14] |
| *zrt101Δ/Δ* | BWP17 | *zrt101::HIS1/zrt101::ARG4* +CIp10 | [27] |
| *zrt2Δ/Δ* | BWP17 | *zrt2::HIS1/zrt2::ARG4* +CIp10 | [28] |
| *zrc1Δ/Δ* | BWP17 | *zrc1::HIS1/zrc1::ARG4* +CIp10 | [28] |
| *pra1Δ/Δ* | BWP17 | *pra1::HIS1/pra1::ARG4* +CIp10 | [27] |
| *zrt101ΔΔ/zrt2ΔΔ* | BWP17 | *zrt1::HIS1/zrt1::ARG4 zrt2::FRT/zrt2::FRT* +CIp10 | [28] |
| *cht2Δ/Δ* | SN125 | *cht2::HIS1/cht2::LEU2* | [66] |

ml Holotransferrin (Calbiochem, Merck), and 1% non-essential amino acids (Gibco, Thermo Fisher Scientific) at 37˚C with 5% $CO_2$ for no longer than 15 passages. C2BBe1 cells were seeded in 6-well plates at a concentration of $5\times10^5$ cells/well for RNA isolation. C2BBe1 and HT29-MTX cells were seeded in 96-well plates and transwell inserts (polycarbonate membrane with 5 μm pores; Corning) at a 70:30 ratio (C2BBe1:HT29-MTX) and a total concentration of $2\times10^4$ cells/well or insert for damage and translocation assays, respectively. For adhesion and invasion assays, C2BBe1 and HT29-MTX cells were seeded in 24-well plates with coverslips at a 70:30 ratio (C2BBe1:HT29-MTX) at a total concentration of $1\times10^5$ cells/well. All well plates and transwell inserts were coated with collagen I (10 μg/ml for 2 h at room temperature; Thermo Fisher Scientific) and maintained for 12 d at 100% confluency for differentiation with regular medium exchange before infection. Just prior to infection with *C. albicans*, the medium was removed and fresh DMEM without FBS, Holotransferrin, or 1% non-essential amino acids was added to the cells.

## RNA isolation

C2BBe1 cells were cultured for 12 d in collagen-coated 6-well plates. To prepare samples for dual-RNA sequencing and qPCR, the differentiated intestinal epithelial cells (IECs) were infected with $2\times10^6$ *Candida* cells/well and incubated at 37˚C and 5% $CO_2$ for 0.75, 3, 6, 12, or 24 h in serum-free DMEM medium. For qPCR samples with added zinc, the fungal and C22Be1 cells were co-incubated in serum-free DMEM with 25 μM $ZnSO_4$ for 24 h. In parallel, C2BBe1 and *C. albicans* cells were cultured individually in 6-well plates in serum-free DMEM medium. *C. albicans* cells from the O/N cultures were collected as uninfected controls for the fungus (0 h *C. albicans*), and uninfected IECs were harvested as well (0 h IECs). At each respective time point, the medium was removed, unadhered *Candida* cells were washed away, and 650 μl of RLT buffer (RNeasy Minikit; Qiagen) was added to each well. The plates were frozen with liquid nitrogen and thawed at room temperature (RT). Each well was scraped and the suspensions were centrifuged for 8 min (20,000×*g*) to pellet the *Candida* cells. The supernatant was removed and the host RNA was isolated using the RNeasy Minikit (Qiagen) according to the manufacturer's instructions. The fungal RNA was isolated from the pellet using a freeze-thaw method described previously [69]. RNA concentrations were measured using a NanoDrop 1000 Spectrophotometer (Thermo Fisher Scientific) and the quality was controlled using a 2100 Bioanalyzer (Agilent Technologies).

Fungal RNA used for q-RT-PCR was isolated as described previously with minor modifications [69]. Briefly, fungal pellets were resuspended in AE buffer (50 mM Na-acetate pH 5.3, 10 mM EDTA) with 0.5% SDS and transferred to screw cap tubes with acid-washed glass beads. An equal volume of phenol/chloroform/isoamylalcohol (25:24:1) was added and the fungal cells were lysed using a FastPrep-24 5GMP Biomedicals. AE buffer was added and the samples were centrifuged for 10 min (20,000×*g*, 4˚C). The aqueous phase was transferred to a PLG-heavy tube (QuantaBio), re-extracted with an equal volume of phenol/chloroform/isoamylalcohol, and centrifuged for 5 min (20,000×*g*, 4˚C). The RNA was then precipitated overnight at -20˚C with pure ethanol and sodium acetate. The RNA concentration was measured using a NanoDrop 1000 Spectrophotometer (Thermo Fisher Scientific) and samples were stored at -70˚C.

## Fungal gDNA isolation

Differentiated C2BBe1:HT29-MTX cells in 24-well plates were infected with $4\times10^5$ *C. albicans* cells and incubated at 37˚C with 5% $CO_2$ for 24 h in serum-free DMEM with or without 25 μM $ZnSO_4$. The medium was removed, the host cells were lysed with 350 μl RLT buffer

(RNeasy Minikit: Qiagen), and the fungal cells were spun down. Fungal gDNA was isolated as previously published with the following modifications [70]. The fungal pellet was washed with water once. Equal volumes of lysis buffer (2% Triton X-100; 1% SDS; 100 mM NaCl; 10 mM Tris, pH 8; 1 mM EDTA) and phenol:chloroform:isoamylalcohol (25:24:1) were added and the samples were transferred to tubes with acid-washed glass beads. The fungal cells were disrupted using a FastPrep-24 5G (MP Biomedicals). Half the total volume of TE buffer (10 mM Tris; 1 mM EDTA; pH 7.5) was added. Nucleic acid was precipitated from the aqueous phase and resuspended in 400 µl TE buffer following RNase A treatment (10 mg/ml; Sigma) for 30 min at 37˚C. gDNA was precipitated in pure ethanol, washed and resuspended in 25 µl nuclease-free water.

## cDNA synthesis and q-RT-PCR

Isolated RNA (1µg) was treated with DNase I (Baseline-ZERO DNase, Lucigen) according to the manufacturer's instructions and then transcribed into complementary DNA using 50µM of oligo (dT)$_{12\text{-}18}$ primer (Invitrogen), 150 U of Superscript III Reverse Transcriptase (Invitrogen), and 10 U RNase OUT (Invitrogen). The cDNA was then diluted 1:5 and used for q-RT-PCR with the GoTaq qPCR Master Mix (Promega) in a CFX96 thermocycler (BioRad). The expression levels were normalized against the *C. albicans ACT1* gene.

For fungal gDNA samples, *EFB1* was amplified from 1µl of template with qPCR Master Mix (Promega) in a CFX Opus Real-Time PCR System (BioRad). All primers used are listed in Table 2. The relative gDNA for each sample was calculated relative to the untreated WT (BWP17) for the respective biological replicate.

## RNA sequencing and transcriptional analysis

Individual *C. albicans* and human C2BBe1 samples were combined for dual-RNA sequencing for uninfected isolated samples for time course data. Both *C. albicans* and *Homo sapiens* reads were present in infected samples. C2BBe1 reads were present in infection data for mutant vs. WT comparison.

Library preparation and RNA sequencing were carried out at Eurofins Genomics GmbH (Ebersberg, Germany) using the Illumina HiSeq 2500 platform (time course analysis) or

**Table 2. Primers used for q-RT-PCR.**

| Name | Sequence 5' to 3' | Gene |
|---|---|---|
| ZRT101 fwd | TCGAAGGTTTGGCTTTGTCT | *ZRT101* |
| ZRT101 rev | CTCATGAGCAACATTCCCAA | *ZRT101* |
| ZRT2 fwd | CAACTACCAATTGGGCCAGA | *ZRT2* |
| ZRT2 rev | GCTCCCCAACACATGACAAA | *ZRT2* |
| ZRC1 fwd | TTTAGTACGTAAAGCCCTGAG | *ZRC1* |
| ZRC1 rev | TCTTGGTCTTGGTCTTGTTCT | *ZRC1* |
| PRA1 fwd | CATTACGCTGACACTTATGAGG | *PRA1* |
| PRA1 rev | ATGTGTGTGGCAATGCAGGT | *PRA1* |
| ACT1 fwd | TCAGACCAGCTGATTTAGGTTTG | *ACT1* |
| ACT1 rev | GTGAACAATGGATGGACCAG | *ACT1* |
| ZRT3 fwd | AGGGGGATTATCTCAACCTTT | *ZRT3* |
| ZRT3 rev | CCACCAAATGAAACACTACTACC | *ZRT3* |
| EFB1 fwd | CGAAATGGAAGGTTTGACTTGG | *EFB1* |
| EFB1 rev | ACAGCAGCTTGTAAGTCATCC | *EFB1* |

Illumina HiSeq 4000 platform (*C. albicans* mutant to WT comparisons). For library preparation, after poly(A) enrichment, mRNA was fragmented and random-primed cDNA synthesis was performed followed by adaptor ligation and adaptor-specific PCR amplification. Single sequence reads of 50 bp were produced.

Preprocessing of raw reads including quality control and gene abundance estimation was done with the GEO2RNaseq pipeline (v0.100.3) [71] in R (version 3.6.3). Quality analysis was done with FastQC (v0.11.5) before and after trimming. Read-quality trimming was done with Trimmomatic (v0.36). Reads were rRNA-filtered using SortMeRNA (v2.1) with a single rRNA database combining all rRNA databases shipped with SortMeRNA. Reference annotation was created by extracting and combining exon features from corresponding annotation files. Reads were mapped against the joined reference genomes of *H. sapiens* (Ensembl_GRCh38) and *C. albicans* (C_albicans_SC5314_version_A22) using HiSat2 (v2.1.0, single end mode). Gene abundance estimation was done with featureCounts (v2.0.1) in single-end mode with default parameters. MultiQC version 1.7 was finally used to summarize and assess the quality of the output of FastQC, Trimmomatic, HiSat, featureCounts and SAMtools. The count matrices with gene abundance data without and with median-of-ratios normalization (MRN) were extracted. Raw files are accessible under the Gene Expression Omnibus accession number GSE237496 (https://www.ncbi.nlm.nih.gov/geo/query/acc.cgi?acc=GSE237496) [72].

Differential gene expression was analyzed using GEO2RNaseq. Pairwise tests were performed using four statistical tools (DESeq2 v1.26.0) to report p values and multiple testing corrected p values using the false-discovery rate method q = FDR(p). Mean MRN values were computed per test per group including corresponding $\log_2$ of fold-changes. Gene expression differences were considered significant if they were reported significant by DESeq2 (q $\leq$ 0.05) and $|\log_2(\text{fold-change[MRN based]})| \geq 1$.

Functional enrichment analysis was performed using overrepresentation analysis. Gene ontology (Biological process) for *C. albicans* was performed by parsing data gene lists to the web application of FungiDB for gene annotation and subsequent analysis using default parameters (https://fungidb.org) [73]. Enrichment for human reads against KEGG pathways were computed using g:Profiler from within R (package gprofiler2, v0.2.1). Enriched terms were discarded if background size was below 3 or above 1000 genes. A bash script was used to call the Revigo web app for removing redundant terms (reduction factor = 0.5, removal of obsolete terms = yes) [74]. KEGG terms were further filtered to exclude coincidental significant hits that were not relevant for the current study (virus-related categories, "Organismal systems" except "Immune system", "Human diseases", "Drug development"). Programming code and data necessary to generate plots shown in this manuscript were deposited at Github: https://github.com/SchSascha/Cal_Translocation.

## Quantification of adhesion, invasion, and hyphal length

Differentiated C2BBe1:HT29-MTX cells in 24-well plates with glass coverslips were infected with $1\times10^5$ *C. albicans* cells and incubated at 37˚C with 5% $CO_2$ for 1 h (adhesion) or 6 h (invasion). For adhesion, samples were fixed with 4% formaldehyde and washed twice with PBS. Adherent *C. albicans* cells were stained with 10 µg/ml Calcofluor white (CFW) (Fluorescent Brightener, Sigma-Aldrich) in 100 mM TRIS-HCl (pH 9.5) for 30 min at RT. After washing with water, samples were mounted on a glass slide with mounting medium (ProLong Gold Antifade, Invitrogen) and imaged. For each replicate, 12 representative images were counted and the mean number of adherent cells in a defined area was calculated from four independent replicates. For quantification of invasion, extracellular *C. albicans* hyphae were stained with CFW as stated above. The host cells were then permeabilized with 0.5% Triton X-100 for 5

min and washed with PBS. Intracellular *C. albicans* hyphae were first labelled with 20–25 μg/ml rabbit anti-*C. albicans* antibody (Acris) for 3 h at 4˚C. After washing with PBS, 4 μg/ml goat anti-rabbit antibody conjugated to AlexaFluor 488 was added and incubated at 37˚C for 1 h. The samples were washed with PBS and mounted as mentioned above. The number of invasive hyphae was determined from 100 hyphae counted per sample. All samples were imaged using an Axio Observer fluorescence microscope (Zeiss). To quantify hyphal length, $2\times10^4$ *C. albicans* cells were seeded per well in a 96-well plate in DMEM and incubated for 3 h at 37˚C with 5% $CO_2$. Images were taken using a Cell Discoverer 7 microscope (Zeiss) and the hyphal length was measured using Zeiss Zen3.4 (blue edition).

## Quantification of host cytotoxicity (LDH assay)

Differentiated C2BBe1:HT29-MTX cells in 96-well plates were infected with $8\times10^4$ *C. albicans* cells and incubated at 37˚C with 5% $CO_2$ for 24 h. Epithelial damage was quantified by release of lactate dehydrogenase (LDH) from IECs. LDH concentrations were measured using a cytotoxicity detection kit (Roche) according to the manufacturer's instructions and LDH from rabbit muscle (Roche) was used to generate a standard curve. The baseline LDH concentration of uninfected and untreated IECs was subtracted and the corrected LDH release is shown as a concentration (ng/ml) present in the supernatant unless otherwise stated.

## *In vitro* translocation assay

Differentiated C2BBe1 cells or a mix of C2BBe1:HT29-MTX cells in transwell inserts were infected with $1\times10^5$ *C. albicans* cells and incubated at 37˚C with 5% $CO_2$. Supernatant from the upper compartment was removed 24 hpi and used for LDH measurement as described above. The lower compartment was treated with 20 U/ml zymolyase (Amsbio) for 2 h at 37˚C and 5% $CO_2$ to detach translocated hyphae. The zymolyase-treated hyphae were then plated on YPD agar, incubated at 30˚C for 2 d, and the colony forming units (CFUs) were counted.

For experiments with NFκB inhibition, C2BBe1 cells were treated with DMEM with 2.5 μM 6-Amino-4-(4-phenoxyphenylethylamino)quinazoline (QNZ) (EVP4593; Sigma-Aldrich) or either 2.5 or 5 μM N-(6-benzoyl-1H-benzo[d]imidazol-2-yl)-2-(1-(thieno[3,2-d]pyrimidin-4-yl)piperidin-4-yl)thiazole-4-carboxamide (SC75741) (MedChemExpress) in the lower compartment for 1 h prior to infection at 37˚C with 5% $CO_2$. IECs were then infected as above. After 24 h the trans-epithelial electrical resistance (TEER) was measured using a volt-ohm meter (EVOM2, World Precision Instruments), supernatants from the upper compartment were collected for LDH measurements, the lower compartment was treated as above to measure CFUs, and the transwell membranes were removed for staining. IECs were fixed with Histofix (Roth) for 10 min at 37˚C then washed with PBS. The cells were permeabilized with 0.5% TritonX-100 for 10 min at RT and washed with PBS. Cells were blocked with 2% BSA for 10 min at 37˚C, washed with PBS, and incubated with a primary mouse anti-E-cadherin antibody (10 μg/ml) (BD Biosciences) for 4 h at 4˚C, then washed with 2% BSA. Cells were then stained with a secondary goat anti-mouse antibody conjugated to AlexaFluor 488 (10 μg/ml) (Invitrogen). Samples were washed with PBS, mounted on glass slides, and images with an Axio Observer fluorescence microscope (Zeiss). Samples were prepared from 3 independent biological replicates and 3 images were taken per strain per condition for each replicate. Images were randomized and scored by a blinded observer. Scores ranged from 1 to 5, with a score of 1 representing a sample with consistent, organized staining of E-cadherin at cell-cell borders throughout the sample. A score of 5 was assigned to samples with no visible staining of E-cadherin at cell-cell junctions. For experiments using the pan-caspase inhibitor (Z-VAD), host

cells in transwell inserts were treated for 1 h prior to infection with 25 μM of the inhibitor alone or in combination with 2.5 μM of QNZ as described above.

For experiments with added zinc, the number of fungal cells was adjusted in either DMEM or DMEM + 25 μM $ZnSO_4$. Differentiated C2BBe1:HT29-MTX cells in transwell inserts were infected as above, with either DMEM or DMEM + 25 μM $ZnSO_4$ in the transwell insert and DMEM in the lower compartment. After 24 h incubation at 37˚C with 5% $CO_2$, the translocated fungal CFUs were plated as described above.

## Immunoblotting

For detection of junction proteins with and without NFκB inhibition, C2BBe1 cells were infected as mentioned for the *in vitro* translocation assay with and without the NFκB inhibitor QNZ and incubated for 24 h at 37˚C with 5% $CO_2$. The transwell membranes were then removed, the host cells were scraped off, and spun down for 5 min at 4˚C at 500×g. The pellets were lysed in RIPA lysis buffer (VWR) supplemented with a protease inhibitor mix (Roche) and Benzonase (Millipore) and chilled for 15 min on ice. The samples were spun down for 15 min at 4˚C at 18,000×*g*. The supernatants were collected and stored at -70˚C. The protein concentrations of total protein extracts were measured by a Lowry protein assay kit (Bio-Rad Laboratories GmbH). 25 μg of each sample was used for a standard 12% SDS polyacrylamide gel electrophoresis (SDS-PAGE) at 30 mA and 200 V. Therefore, lysates were supplemented with SDS sample buffer (50 mM Tris/HCl, 2% (v/v) glycerol, 2% (v/v) β-mercaptoethanol, 1.6% (w/v) SDS, 0.004% (w/v) Serva Blue G-250, pH 6.8) and boiled for 10 min at 95˚C. After SDS-PAGE with subsequent blotting of the samples onto an Amersham Protran 0.45 μm nitrocellulose membrane at 1.8 mA/cm$^2$ and 25 V, blots were incubated with 5% (w/v) milk dissolved in TBS (50 mM Tris/HCl, 140 mM NaCl, pH 7.2) for 1 h at room temperature. For specific, immunologic labeling, primary antibodies against E-Cadherin (1:1000, MAB1838, R&D Systems), Claudin-1 D5H1D (1:1000, 13255, Cell Signaling Technology), and GAPDH D16H11 (1:1000, 5174, Cell Signaling Technology) were used in 5% (w/v) milk/TBS for 3 h at room temperature or 16–48 h at 4˚C. After 3 times washing in TBS-T (TBS supplemented with 0.05% (v/v) Tween-20), membranes were incubated for 1.5–3 h with IRDye 800CW donkey anti-mouse or donkey anti-rabbit (1:10000, 925–32212 and 925–32213, LI-COR GmbH) as secondary antibodies, respectively. Immunological detections were carried out using an Odyssey M imaging system (LI-COR GmbH). Blots were analyzed by Image Studio Lite Version 5.0 (LI-COR GmbH).

For detection of immune signaling proteins, C2BBe1 cells were seeded in 6-well plates and infected with WT (BWP17) and *ece1*Δ/Δ *C. albicans* as described above for RNA isolation experiments. After 6 h of infection, tissue culture plates were placed on ice, the medium was removed, and the cells were washed with ice-cold PBS. Cells were lysed with 120 μL of RIPA buffer (25 mM Tris-HCl pH 7.4, 150 mM NaCl, 1% Nonidet P-40 (NP-40), 1 mM EDTA and 5% glycerol) supplemented with protease and phosphatase inhibitors (1:100 dilution) (Sigma Aldrich). Adherent cells were then scraped, transferred into pre-cooled microfuge tubes and incubated on ice for 30 min. Lysates were clarified by centrifugation at 13,300×*g* at 4˚C for 10 min. The protein extract concentration was measured using a bicinchoninic acid assay (BCA) (Thermo Fisher Scientific) according to the manufacturer's instructions. Proteins were resolved by electrophoresis on 20% SDS-PAGE gels. Following electrophoresis, proteins were transferred onto nitrocellulose membranes (Bio-Rad). Membranes were blocked in 1× Tris-buffered saline (TBS; Severn Biotech) containing 0.001% Tween 20 (Acros Organics) and 5% skimmed milk powder (Sainsbury's). After washing once with TBST, membranes were incubated with primary antibody (Table 3) with gentle agitation overnight at 4˚C. The following

**Table 3. Detection antibodies for immune signaling proteins.**

| Antibody | Species | Dilution | Company | Catalogue Number |
|---|---|---|---|---|
| α-actin (clone C4) | Mouse | 1:10,000 | Merck Millipore | MAB1501 |
| p-DUSP1/MKP1 (S359) | Rabbit | 1:1,000 | Cell Signaling | 2857 |
| Total p38 | Rabbit | 1:1,000 | Cell Signaling | 8690 |
| Total EGFR | Rabbit | 1:1,000 | Cell Signaling | 4267 |
| Total Akt | Rabbit | 1:1,000 | Cell Signaling | 9272 |
| Total-EphA2 | Rabbit | 1:1,000 | Cell Signaling | 6997 |
| p-p38 (T180/Y182) | Rabbit | 1:1,000 | Cell Signaling | 4511 |
| p-Akt (Ser473) | Rabbit | 1:1,000 | Cell Signaling | 9271 |
| p-EGFR (Tyr1068) | Rabbit | 1:1,000 | Cell Signaling | 3777 |
| p-EphA2 (S897) | Rabbit | 1:1,000 | Cell Signaling | 6347 |
| c-Fos | Rabbit | 1:1,000 | Cell Signaling | 2250 |
| Peroxidase-conjugate AffiniPure anti-mouse IgG | Goat | 1:20,000 | Jackson ImmunoReasearch | 115-035-062 |
| Peroxidase-conjugate AffiniPure anti-rabbit IgG | Goat | 1:20,000 | Jackson ImmunoReasearch | 115-035-003 |

day, membranes were washed 3 times for 5 min with TBST. Membranes were subsequently incubated with rabbit or mouse secondary antibody (Thermo Fisher Scientific) for 1 h at room temperature and then washed 6 times for 5 min with TBST. Finally, the proteins were detected using Immobilon western Chemiluminescent HRP Substrate (Merck Millipore) and developed with an Odyssey Fc Imaging System (LI-COR). Human α-actin was used as a loading control.

## Statistical analyses

All experiments were performed with at least three biological replicates. Data were analyzed using Prism 9.4 (GraphPad Software). All data used to generate the graphs is provided as source data (S3 Table).

## Supporting information

**S1 Fig.** *C. albicans* **zinc-associated genes are more highly expressed in the absence of IECs.** Expression of the genes involved in zinc transport (*ZRT101*, *ZRT2*, *ZRT3*, *ZRC1*), zinc scavenging (*PRA1*), and regulation of zinc acquisition genes (*ZAP1*). Log$_2$(fold-change) compares infected samples to the yeast pre-culture conditions on the left and medium-only samples to the yeast pre-culture conditions on the right. Asterisks indicate time points with significantly expression changes (DESeq2 $p < 0.05$).
(TIF)

**S2 Fig. Loss of** *ZRC1* **significantly impairs hyphal growth in** *C. albicans.* Loss of *ZRT101*, *ZRT2*, or *PRA1* did not significantly impact hypha formation. However, loss of *ZRC1* significantly decreased hyphal length in *C. albicans* in cell culture medium after 6 h. All values are shown as the mean with standard deviation. Data were compared using a one-way ANOVA with a post-hoc Dunnett's multiple comparisons test. Statistical significance: *, $P \leq 0.05$.
(TIF)

**S3 Fig. Loss of** *ECE1* **has no effect on the transcriptional zinc starvation response in medium only.** Fold change in gene expression in *C. albicans* WT (BWP17) and *ece1*Δ/Δ strains during incubation in cell culture medium for (A) *ZRT101*, (B) *ZRT2*, (C) *PRA1*, (D) *ZRT3*, and (E) *ZRC1*. All values are shown as the mean with standard deviation.
(TIF)

**S4 Fig. Addition of exogenous zinc alleviates the transcriptional zinc starvation response in the absence of *ECE1* but does not affect host-cell damage, fungal translocation, or fungal load.** (A) Host-cell damage in the absence or presence of 25 μM exogenous ZnSO$_4$ for the WT mutant (same data presented in Fig 2D) and the *ece1*Δ/Δ strain. (B) Fungal translocation of the WT(BWP17) and *ece1*Δ/Δ strains with or without the addition of 25 μM ZnSO$_4$. (C) Relative quantification of fungal gDNA during infection of IECs. All samples are compared to the WT (BWP17) without added zinc. (D) Fold change in normalized gene expression of the *ece1*Δ/Δ strain compared to WT(BWP17) at 24 h during infection of IECs with addition of 25 μM ZnSO$_4$. Gene expression was normalized to *ACT1* as a housekeeping gene. All values are shown as the mean with standard deviation.
(TIF)

**S5 Fig. Transcriptional response of IECs to infection with non-damaging and non-filamentous *C. albicans*.** Genes differentially expressed when comparing the non-damaging *ece1*Δ/Δ and the non-filamentous *efg1*ΔΔ/*cph1*ΔΔ strains to their respective WT strains (*ece1*Δ/Δ compared to WT (BWP17) and *efg1*ΔΔ/*cph1*ΔΔ compared to WT (SC5314)). The data are shown as the Log$_2$(fold-change) of infected cells with the different strains compared to uninfected IECs. Asterisks indicate genes with statistically significant differences in expression (DESeq2 $p < 0.05$).
(TIFF)

**S6 Fig. Western blot detection of immune signaling proteins for IECs infected with WT and *ece1*Δ/Δ.** (A) Confluent, differentiated C2BBe1 cells were infected with WT (BWP17) and *ece1*Δ/Δ *C. albicans* for 6 h and the protein content was sampled. Proteins involved in the damage response of oral epithelial cells were detected with *ACT1* serving as a control. n.d. = not determined. (B) Protein levels normalized to actin. For p38, MKP1, EPHA2, EGFR, and AKT the normalized protein level for the phosphorylated protein is presented relative to the total respective protein level.
(TIF)

**S7 Fig. DMSO vehicle control has no effect on host damage or translocation of *C. albicans*.** C2BBe1 cells were treated with a DMSO vehicle control and infected with the WT or *ece1*Δ/Δ *C. albicans* strains. There were no significant changes in (A) host cell damage or (B) fungal translocation. All values are shown as the mean with standard deviation.
(TIF)

**S8 Fig. Treatment of IECs with another NFκB inhibitor increases host cell damage, loss of barrier integrity, and fungal translocation.** (A) Inhibition of NFκB activation using the NFκB inhibitor SC75741 at concentrations of either 2.5 or 5 μM increased the damage of WT (BWP17) *C. albicans*, but not for the *ece1*Δ/Δ strains. LDH release was adjusted by subtracting the release from uninfected and untreated host cells. (B) NFκB inhibition using either concentration also decreased the barrier integrity during infection with both WT and *ece1*Δ/Δ strains. (C) Fungal translocation was significantly increased during infection with both WT and *ece1*Δ/Δ *C. albicans* upon inhibition of NFκB using both concentrations of SC75741. These results match those obtained with the high-affinity NFκB inhibitor quinazoline. All values are shown as the mean with standard deviation. Host-cell damage (A), barrier integrity (B), and fungal translocation (C) data were compared using a one-way ANOVA with a post-hoc Šidák's multiple comparisons test. Statistical significance: *, $P \leq 0.05$; **, $P \leq 0.01$; ***, $P \leq 0.001$; ****, $P \leq 0.0001$.
(TIF)

**S9 Fig. Treatment with a pan-caspase inhibitor does not prevent increased virulence upon inhibition of NFκB activation.** Treatment with a pan-caspase inhibitor (Z-VAD) had no significant effect on (A) host-cell damage, (B) barrier integrity, or (C) fungal translocation when used alone or in combination with the NFκB inhibitor quinazoline (QNZ). All values are shown as the mean with standard deviation. Host-cell damage (A), barrier integrity (B), and fungal translocation (C) data were compared using a one-way ANOVA with a post-hoc Šidák's multiple comparisons test. Statistical significance: *, $P \leq 0.05$; ***, $P \leq 0.001$; ****, $P \leq 0.0001$.
(TIF)

**S10 Fig. Western blot detection of cell-cell junction proteins for IECs during *C. albicans* infection and NFκB inhibition.** Confluent, differentiated C2BBe1 cells were infected with WT *ece1Δ/Δ*, and *efg1ΔΔ/cph1ΔΔ C. albicans* for 24 h. Samples were either untreated or treated with an NFκB activation inhibitor (QNZ) and the protein content was sampled. (A) Proteins that make up tight and adherens junctions (E-cadherin and claudin-1) were detected with GAPDH serving as a control. (B) Claudin-1 protein levels normalized to GAPDH and presented relative to levels in untreated C2BBe1 cells. QNZ treatment further increased degradation of claudin-1, even during infection with *efg1ΔΔ/cph1ΔΔ* and *ece1Δ/Δ*. All values are shown as the mean with standard deviation.
(TIF)

**S1 Table. Differential expression data during *C. albicans*-interaction.** Infection-specific DEGs of *C. albicans* and the host at different time points of infection and host DEGs during infection with either the *ece1Δ/Δ* or *efg1ΔΔ/cph1ΔΔ* strains compared to infection with the WT as stated in the Description sheet.
(XLSX)

**S2 Table. Overrepresentation analysis of DEGs.** Enrichment analysis for *C. albicans* and host DEGs from both the IEC infection time course and mutant comparison RNA sequencing datasets as stated in the Description sheet.
(XLSX)

**S3 Table. Source data.** All data used to generate graphs for main text and supplementary figures.
(XLSX)

## Acknowledgments

We would like to thank all members of the Microbial Pathogenicity Mechanisms department for their valuable feedback and fruitful discussions, in particular Simone Schiele and Julia Mantke for their technical assistance in performing the hyphal length assay and NFκB ELISA, respectively. We thank Thomas Beder, Thomas Wolf, and Sascha Brunke for their initial analyses of the transcriptomic data. We thank Ilse Jacobsen and Nicole Engert-Ellenberger for their help with taking samples for the dual-RNA sequencing. We additionally thank Sascha Brunke for a critical reading of the manuscript.

## Author Contributions

**Conceptualization:** Jakob L. Sprague, Gianni Panagiotou, Julian R. Naglik, Duncan Wilson, Lydia Kasper, Bernhard Hube.

**Data curation:** Jakob L. Sprague, Tim B. Schille, Stefanie Allert, Verena Trümper, Adrian Lier, Peter Großmann, Emily L. Priest, Antzela Tsavou, Sascha Schäuble, Lydia Kasper.

**Formal analysis:** Tim B. Schille, Verena Trümper, Peter Großmann, Sascha Schäuble.

**Funding acquisition:** Bernhard Hube.

**Investigation:** Jakob L. Sprague, Lydia Kasper, Bernhard Hube.

**Methodology:** Jakob L. Sprague, Tim B. Schille, Stefanie Allert, Verena Trümper, Adrian Lier, Emily L. Priest, Antzela Tsavou, Sascha Schäuble.

**Project administration:** Jakob L. Sprague, Gianni Panagiotou, Julian R. Naglik, Duncan Wilson, Sascha Schäuble, Lydia Kasper, Bernhard Hube.

**Resources:** Bernhard Hube.

**Software:** Peter Großmann, Sascha Schäuble.

**Supervision:** Lydia Kasper, Bernhard Hube.

**Validation:** Jakob L. Sprague.

**Writing – original draft:** Jakob L. Sprague, Lydia Kasper.

**Writing – review & editing:** Jakob L. Sprague, Tim B. Schille, Stefanie Allert, Verena Trümper, Adrian Lier, Peter Großmann, Emily L. Priest, Antzela Tsavou, Gianni Panagiotou, Julian R. Naglik, Duncan Wilson, Sascha Schäuble, Lydia Kasper, Bernhard Hube.

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
