## [Decision Letter · Decision Letter 0]

18 Sep 2023

Dear Prof. Hube,

Thank you very much for submitting your manuscript "Candida albicans translocation through the intestinal epithelial barrier is promoted by fungal zinc acquisition and limited by NFκB-mediated barrier protection" for consideration at PLOS Pathogens. As with all papers reviewed by the journal, your manuscript was reviewed by members of the editorial board and by several independent reviewers. In light of the reviews (below this email), we would like to invite the resubmission of a significantly-revised version that takes into account the reviewers' comments.

I am returning your manuscript with three reviews. The reviewers were agreed that the topic was interesting, but each identified some critical issues that need to be addressed. After reading the reviews and looking at the manuscript, I recommend Major Revision based on the critiques. I am sorry I cannot be more positive at the moment, however we are looking forward to receiving your revision. With a lot of work, the manuscript will be suitable for a resubmission, if you so wish to do so. I anticipate that we will send your paper back to the reviewers upon resubmission.

During the revision, please pay particular attention to the following reviewer suggestions and give them due consideration.

• All three reviewers asked for additional controls and care in the interpretation of the dual RNAseq, especially with the conclusion that the fungal signature did not change.

• The experiments on NF-κB require additional, orthogonal, methods to support the conclusions. QNZ alone is not sufficient.

We cannot make any decision about publication until we have seen the revised manuscript and your response to the reviewers' comments. Your revised manuscript is also likely to be sent to reviewers for further evaluation.

Sincerely,

Teresa R O'Meara, Ph.D.

Guest Editor

PLOS Pathogens

Michal Olszewski

Section Editor

PLOS Pathogens

Kasturi Haldar

Editor-in-Chief

PLOS Pathogens

orcid.org/0000-0001-5065-158X

Michael Malim

Editor-in-Chief

PLOS Pathogens

orcid.org/0000-0002-7699-2064

I am returning your manuscript with three reviews. The reviewers were agreed that the topic was interesting, but each identified some critical issues that need to be addressed. After reading the reviews and looking at the manuscript, I recommend Major Revision based on the critiques. I am sorry I cannot be more positive at the moment, however we are looking forward to receiving your revision. With a lot of work, the manuscript will be suitable for a resubmission, if you so wish to do so. I anticipate that we will send your paper back to the reviewers upon resubmission.

During the revision, please pay particular attention to the following reviewer suggestions and give them due consideration.

• All three reviewers asked for additional controls and care in the interpretation of the dual RNAseq, especially with the conclusion that the fungal signature did not change.

• The experiments on NF-κB require additional, orthogonal, methods to support the conclusions. QNZ alone is not sufficient.

Reviewer's Responses to Questions

**Part I - Summary**

Reviewer #1: This manuscript by Sprague et al employed the dual RNAseq strategy to explore the pathogen and host factors during intestinal epithelial cell infection by C. albicans. The authors plotted the differentially expressed genes in both fungal and host cell compartments and found zinc acquisition and NFκB pathway as key processes, respectively. Although dual RNAseq is a powerful tool in studying host-pathogen interactions, the experimental design has deviated from the scenario of Candida intestinal translocation, which hampers the data value.

Reviewer #2: This study by Sprague et al analyzes the fungal and host pathways involved in infection, invasion, damage, and translocation of C. albicans through cultured human intestinal epithelial cells (C2BBe1). Several approaches were used to interrogate this interaction; first, a dual RNA-sequencing approach was used to profile both fungal and epithelial transcriptional changes throughout the infection. For C. albicans, gene expression was compared between fungi cultured in media alone vs infection of IECs. This analysis revealed a number of transcriptional changes specifically associated with IEC infection, which were mainly differences in transcripts associated with nutrient acquisition. The study focuses specifically on alterations in the zinc acquisition and detoxification pathways for functional analysis. Genes encoding zinc acquisition transporters and zincophores were significantly decreased in expression during IEC infection compared to media only conditions, suggesting that C. albicans are more zinc limited in media alone compared to with IECs. Disruption of genes encoding zinc transporters, ZRT101 and ZRT2, together lead to a significant decrease in cytotoxicity, which was rescued by the addition of exogenous zinc. This decrease in cytotoxicity was likely due to a significant growth defect of the zrt101∆/∆ zrt2∆/∆ during infection, which was rescued by addition of zinc. The authors test whether the cytolytic toxin, Candidalysin, alleviates the zinc starvation response, likely through the release of IEC zinc. This analysis shows that specifically during co-culture with IECs does the ece1∆/∆ strain show a significant increase in expression of zinc starvation-associated genes. The authors then investigate the transcriptional response of the IECs to infection, finding changes that are largely similar to what has been observed in other C. albicans/epithelial interactions studies (though with some distinctions that were not explored in detail here). The study then focuses specifically on the role of NFkB activation in C. albicans damage and translocation. Transcripts associated with NFkB signaling are significantly upregulated at 6+ hours post infection, which coincides with C. albicans invasion and damage of IECs. Additional transcriptional profiling experiments with hyphal deficient and ece1∆/∆ mutants show that NFkB transcripts were activated regardless of hyphal and Candidalysin production. The authors use a NFkB activation inhibitor to functionally interrogate whether NFkB signaling was differentially required for blocking all C. albicans strains from damaging and translocating through the IEC layer. They find that NFkB inhibition exacerbates WT C. albicans cytotoxicity and translocation, but has differential effects on the hyphal deficient and ece1∆/∆ mutants. NFkB inhibition promotes ece1∆/∆ but not the hyphal deficient mutant, translocation. The authors then show that NFkB inhibition leads to loss of cell surface e-cadherin localization in cells coincubated with the ece1∆/∆.

Overall, this is a comprehensive study that reveals important mechanisms behind C. albicans infection and translocation through intestinal epithelial cells. These results are significant given that intestine is an important reservoir for C. albicans, and that transit through the intestinal epithelium is thought to be a main route of invasion that seeds disseminated infections. Efforts to interrogate the role of both C. albicans and IEC pathways are impressive and add to the rigor of the study. I believe this is a useful study for the field of C. albicans/epithelial interactions.

Reviewer #3: Sprague et al use tandem RNA-sequencing to identify a putative mechanism in both CA and the host IEC resulting in translocation of CA across the mucosal barrier and subsequent Candidemia.

For IEC the authors use a CACO2 derivative C2BBe1 seeded with a laboratory reference strain of CA with samples collected on a reasonable timeline to capture early and late events (0, 0.75, 3, 6, 12, and 24h).

Curiously, the transcriptional response in CA over time was largely independent of the presence of host cells. DEG were identified with DESEQ2, and notable for the exception (to the broad pattern above) of zinc-related genes being more prominent in CA co-cultured with IEC as compared to medium-only. This observation was followed-up by a phenotypic study of zrc1Δ/Δ mutant that did not have deficits in adhesion or hyphae but was notable for reduced translocation. Damage (as measured by LDH) could be rescued in some but not all mutants (e.g., zrc1) by exogenous zinc supplementation. The authors then compare wildtype CA grown without IEC to an ece1Δ/Δ strain with IEC, demonstrating a transcriptional pattern in both consistent with Zinc-starvation (lacking in Wt CA with IEC). The authors conclude based on this work that zinc acquisition from candidalysin-mediated IEC injury is required for subsequent invasion.

In contrast, the transcriptional changes in IEC over time demonstrate a clear difference between CA exposed or not, further a temporal response only in CA exposed. The authors go on to note a pattern of gene expression associated with NFkB activation with damage-capable CA strain co-culture, including a colometric assay of NFkB activation.

The authors ultimately present a model in which candidalysin-mediated damage to the host is required to provide the CA with zinc, and this damage (and loss of barrier integrity) are opposed by NFkB and NFkB activated genes in the IEC. This core model seems supported by their experiments, with the story more compelling and comprehensive from the CA perspective. For the host side, the authors uncover damage-dependent and damage-independent mechanisms. The linkage between NFkB activation and resilience against CA is not fully developed here (and arguably beyond the scope), and the use of immortalized lines rather than (now readily available) organoid-derived IEC are not well noted in the discussion. Nor are these findings well contextualized with the clinical associations with risk for invasive CA infection (steroids, zinc-deficiency in the host). Overall this is a solid story that could use some better contextualization.

**Part II – Major Issues: Key Experiments Required for Acceptance**

Reviewer #1: 1. In the dual RNAseq experiment, before the fungal cells and C2BBe1 cells seeded into the DMEM medium, fungal cells were cultured in the YPD broth. With the transition from the rich medium (YPD) to the nutrient-limit medium (DMEM), the fungal cells would experience a drastic metabolic rewiring, which may affect the sensitivity of RNAseq in detecting differentially expressed genes to host cell. In contrast, host cells were consistently maintained in DMEM environment and the only difference at the start of the experiment was the depletion of serum. This might explain why transcription profiles of fungal compartment were generally similar with or without host cells at each time points. It is challenging to prepare and quantify C. albicans cells in DMEM medium for this experiment since a lot of the cells would develop to hyphae and become clumpy. But the authors should discuss this when explaining the pattern of fungal DEGs and need to be aware of this limitation before claiming “fungal response was largely independent of the presence of host cells” (line108).

2. The authors used C2BBe1 cell line as a surrogate for intestinal barrier and found genes in zinc acquisition pathway differentially expressed during infection. However, such in vitro scenario might not be that relevant to the real world. In the in vitro system, the DMEM media is zinc-free, fungal genes related to zinc acquisition are highly upregulated and very sensitive to the zinc from those lysed host cells. In contrast, the human intestinal lumen is normally a zinc-rich environment, this is different from other infectious sites such as the oral cavity. In the gut, C. albicans does not need to damage epithelial cells to acquire zinc. Will these fungal genes be differentially expressed during infection in the presence of zinc? A qPCR test would help answer this question.

3. The loss of function of NFκB was only achieved via chemical inhibitor. The authors should address the specificity level of QNZ on inhibiting NFκB, and use another approach (another inhibitor, knock down the gene, etc.) to perform one key experiment (damage or translocation) and show a consistent result.

Reviewer #2: 1) The role for zinc acquisition during IEC infection is interesting. The ece1∆/∆ mutant has increased expression of zinc transporters during infection. Is this a biologically meaningful result? That is, does the ece1∆/∆ mutant have a growth defect compared to WT that is better rescued by the addition of zinc?

2) The discussion around the QZN-treatment experiments are overinterprreted or overstated. The text repeatedly states that QZN has little to no impact on ece1∆/∆ cytotoxicity, but there appears to be an increase in LDH during QZN treatment, which may be biologically significant given the severe defect in ece1∆/∆ without QZN (basically zero cytotoxicity in 2D). I do not agree that the impact of QZN on ece1∆/∆ translocation is completely due to a cytoxicity-independent pathway. If NFkB inhibition promoted fungal translocation in a damage -independent manner, wouldn’t you expect to see increased translocation of the hyphal deficient strain? Why is there no E-cadherin mis-localization with the hyphal-deficient strain?

Reviewer #3: Major Point:

- Figure 3 would ideally include the gene expression data of WT CA without IEC for comparison, as this is a key point in the results and overall model (i.e., that zinc starvation in ece1Δ/Δ is comparable to WT without IEC).

- The legend for Figure 5C does not reflect the data. I.e., for the WT CA the inhibition of NFkB did not have a significant effect on the loss of TEER. The paired result text more accurately reflects the results.

- The e-cadherin results (5E) could use a better figure legend and perhaps some effort at quantification (e.g., scoring of e-cadherin organization by a blinded observer) and more than one replicate per condition.

- The extensive use of an immortalized cell line as a surrogate for IEC responses is both reasonable and a major limitation not noted by the authors in the discussion. Particularly given the readily available organoid-derived human IEC (including commercially available via Millipore), and repeatedly demonstrated differences in immortalized cell line IEC as compared to organoid or native host tissue responses, this limitation should at least be clearly acknowledged.

- The authors note that patient factors are major considerations for eventual risk of invasive CA infection, citing antibiotic exposure. But their own work is poorly contextualized in the discussion. For example, the clinical risks from NFkB inhibition (e.g., steroid use clinically) and poor nutritional states (e.g., zinc deficiency is common in chronic / severe illness) are fitting with the author’s experimental findings. Clincally, the risk of zinc deficiency is well-recongized enough to be the basis of recommended care for pediatric patients in the ICU (e.g., https://pubmed.ncbi.nlm.nih.gov/36329878/)

**Part III – Minor Issues: Editorial and Data Presentation Modifications**

Reviewer #1: 1. To define filamentation-specific and damage-specific response host genes, it is better to have the fungal mutations, ece1Δ/Δ and efg1ΔΔ/cph1ΔΔ, in the same strain background. Although BWP17 is derived from SC5314, there are still genetic differences between the two isolates. The authors can briefly address this when discussing host response genes.

2. The legend of Figure 1B, “Gene ratio represents the proportion of genes within each category that were significantly differentially expressed during infection compared to IECs only.” This figure is about the DEGs in Candida, should be “compared to C. albicans only”.

3. Figure 2D, show individual data points as figure 2A-C and other figure panels, not just the error bar.

4. Figure 2E, image quality needs to be improved. May add arrows or dash line to point out the key information or quantify the hyphal length or growth via imaging software.

Reviewer #2: 1) The change in E-cadherin protein levels are relatively unchanged according to Fig 5D, but are discussed as if there are meaningful differences between the C. albicans strains.

Minor comments:

2) Fig. 1B) legend states: *of a population of unicellular organisms in response to chemical **between organisms. It is unclear what this statement refers to. Please clarify.

3) Also in the figure legend for 1B, it states the transcripts are compared to IECs only – I believe this should be compared to medium only controls?

4) Lines 487-488 state: “The low responsiveness of IECs to C. albicans infection may be due to the normally commensal lifestyle within the gastrointestinal tract “ C. albicans is also commensal colonizer of the oropharyngeal and urogenital tracts correct? I do not think this is relevant to how these different epithelial cells respond to C. albicans.

Reviewer #3: Minor points:

- Figure 1C and Fig 4C depicts log2 fold change. Either in this figure or another figure the actual FPKM / RPKM values should be shown, not just the fold change.

- Figure legends ideally should include the test used (e.g., figure 2, 3, etc do not specify the test used) when a p value is provided.

- The article has some awkward phrasing in English and could use a copy-edit to aid in clarity.

- Figure 4C could use an x-axis label for clarity.

- Figure 4E y-axis label ideally would specify NFkB activity rather than the raw reading. The later is fine for the legend (to explain how activity was estimated) but the former would be clearer for a reader to understand what is shown.

- Ideally, reviewers would be provided access (via a reviewer key) to the raw read data. This can be done while keeping the read data non-public.

PLOS authors have the option to publish the peer review history of their article (what does this mean?). If published, this will include your full peer review and any attached files.

Reviewer #1: No

Reviewer #2: No

Reviewer #3: No
---

## [Decision Letter · Decision Letter 1]

21 Dec 2023

Dear Prof. Hube,

Thank you very much for submitting your revised manuscript "Candida albicans translocation through the intestinal epithelial barrier is promoted by fungal zinc acquisition and limited by NFκB-mediated barrier protection" for consideration at PLOS Pathogens. As with all papers reviewed by the journal, your manuscript was reviewed by members of the editorial board and by several independent reviewers. In light of the reviews (below this email), we would like to invite the resubmission of a significantly-revised version that takes into account the reviewers' comments.

The revised manuscript is a substantial improvement, and we appreciate the addition of new experiments that increased the strength of the manuscript. However, there are a few key experiments that need to be completed. These studies are essential to support the conclusion. We are looking forward to your revised manuscript after you complete these essential studies. Please let me know if additional time is needed to perform these experiments.

We cannot make any decision about publication until we have seen the revised manuscript and your response to the reviewers' comments. 

Sincerely,

Teresa R O'Meara, Ph.D.

Guest Editor

PLOS Pathogens

Michal Olszewski

Section Editor

PLOS Pathogens

Kasturi Haldar

Editor-in-Chief

PLOS Pathogens

orcid.org/0000-0001-5065-158X

Michael Malim

Editor-in-Chief

PLOS Pathogens

orcid.org/0000-0002-7699-2064

Reviewer's Responses to Questions

PLOS authors have the option to publish the peer review history of their article (what does this mean?). If published, this will include your full peer review and any attached files.

Reviewer #1: No

Reviewer #2: No

**Part I - Summary**

Reviewer #2: The study by Sprague et al has been strengthened by additional experiments and qualifying the interpretation of results. Main new experiments include 1) showing that ECE1-dependent regulation of zinc starvation response genes is mostly rescued by the addition of ZnSO4, 2) the addition of a separate NFkB inhibitor verifying ECE1-independent increase in C. albicans translocation through IECs. 3) However, there are 2 issues that must be addressed.

1) The authors argue that a more complex culture system would be required to determine whether zinc starvation causes growth defects or translocation defects in the ece1∆/∆ strain vs the wild type. I do not understand why a complex culture system would be required to explore these phenotypes. For example, is the fungal burden of ece1∆/∆ increased compared to wild type upon ZnSO4-treated samples shown in SI Figure 3.

2) This statement is not supported by the data: “The host-cell damage defects of zinc uptake impaired mutants (lacking ZRT101, ZRT2, or PRA1), but not of a zinc detoxification defective mutant (zrc1Δ/Δ) were rescued by addition of 25 μM ZnSO4, a zinc concentration known to promote C. albicans growth while not being toxic (Fig 2D) (28)” The data shows no apparent difference in LDH release levels between untreated and ZnSO4 treatment for zrt101∆/∆, zrt2∆/∆, or pra1∆/∆. The authors instead are interpreting a lack of statistical difference between LDH release in ZnSO4-treated wild type and mutant strains as evidence of rescued LDH release. A direct comparison of ZnSO4-treated vs untreated for these strains would be necessary to make this statement.

**Part II – Major Issues: Key Experiments Required for Acceptance**

Reviewer #2: 1) The authors argue that a more complex culture system would be required to determine whether zinc starvation causes growth defects or translocation defects in the ece1∆/∆ strain vs the wild type. I do not understand why a complex culture system would be required to explore these phenotypes. For example, is the fungal burden of ece1∆/∆ increased compared to wild type upon ZnSO4-treated samples shown in SI Figure 3.

2) This statement is not supported by the data: “The host-cell damage defects of zinc uptake impaired mutants (lacking ZRT101, ZRT2, or PRA1), but not of a zinc detoxification defective mutant (zrc1Δ/Δ) were rescued by addition of 25 μM ZnSO4, a zinc concentration known to promote C. albicans growth while not being toxic (Fig 2D) (28)” The data shows no apparent difference in LDH release levels between untreated and ZnSO4 treatment for zrt101∆/∆, zrt2∆/∆, or pra1∆/∆. The authors instead are interpreting a lack of statistical difference between LDH release in ZnSO4-treated wild type and mutant strains as evidence of rescued LDH release. However, a direct comparison of ZnSO4-treated vs untreated for these strains would be necessary to make this statement.

**Part III – Minor Issues: Editorial and Data Presentation Modifications**

Reviewer #2: Quantification of SI Fig. 5 would be helpful in visualizing the conclusions of this data.
---

## [Decision Letter · Decision Letter 2]

6 Feb 2024

Dear Prof. Hube,

We are pleased to inform you that your manuscript 'Candida albicans translocation through the intestinal epithelial barrier is promoted by fungal zinc acquisition and limited by NFκB-mediated barrier protection' has been provisionally accepted for publication in PLOS Pathogens.

Should you, your institution's press office or the journal office choose to press release your paper, you will automatically be opted out of early publication. We ask that you notify us now if you or your institution is planning to press release the article. All press must be coordinated with PLOS.

Best regards,

Teresa R O'Meara, Ph.D.

Guest Editor

PLOS Pathogens

Michal Olszewski

Section Editor

PLOS Pathogens

Michael Malim

Editor-in-Chief

PLOS Pathogens

orcid.org/0000-0002-7699-2064

Reviewer Comments (if any, and for reference):

Reviewer's Responses to Questions

**Part I - Summary**

Reviewer #2: The authors addressed all of my concerns.

**Part II – Major Issues: Key Experiments Required for Acceptance**

Reviewer #2: none

**Part III – Minor Issues: Editorial and Data Presentation Modifications**

Reviewer #2: none

PLOS authors have the option to publish the peer review history of their article (what does this mean?). If published, this will include your full peer review and any attached files.

Reviewer #2: No

---

## [Editor Report · Acceptance letter]

16 Feb 2024

Dear Prof. Hube,

We are delighted to inform you that your manuscript, "Candida albicans translocation through the intestinal epithelial barrier is promoted by fungal zinc acquisition and limited by NFκB-mediated barrier protection," has been formally accepted for publication in PLOS Pathogens.

Best regards,

Michael Malim

Editor-in-Chief

PLOS Pathogens

orcid.org/0000-0002-7699-2064